# LucidAtlas: Learning Uncertainty-Aware, Covariate-Disentangled, Individualized Atlas Representations

**Yining Jiao**[1,2]*, **Sreekalyani Bhamidi**[1], **Carlton Zdanski**[1], **Huaizhi Qu**[1],
**Julia Kimbell**[1], **Andrew Prince**[1], **Cameron Worden**[1], **Samuel Kirse**[1],
**Christopher Rutter**[1], **Benjamin Shields**[1], **Jisan Mahmud**[1], **Tianlong Chen**[1],
**Marc Niethammer**[2]
[1] *UNC-Chapel Hill*    [2] *UC San Diego*

*Reviewed on OpenReview: https://openreview.net/forum?id=3FbNwC8ua8*

## Abstract

Interpreting how covariates influence spatially structured biological variation — for example, how pediatric airway geometry changes along the airway and across a growing population — remains a key challenge in developing models suitable for clinical application. We present `LucidAtlas`, a versatile framework for modeling and interpreting spatially varying information with associated covariates. To address the limitations of neural additive models when analyzing dependent covariates, we introduce a marginalization approach that enables accurate explanations of how combinations of covariates shape the learned atlas. `LucidAtlas` integrates covariate interpretation, spatial representation, individualized prediction, population distribution analysis, and out-of-distribution detection into a single interpretable model. We validate its effectiveness on a synthetic spatiotemporal dataset, the OASIS brain volume dataset, and a pediatric airway shape dataset. Our findings underscore the critical role of by-construction interpretable models in advancing scientific discovery. The implementation is publicly available at `https://github.com/ucsdbiag/LucidAtlas`.

## 1 Introduction

Understanding how anatomy varies across a population and with covariates such as age, weight, and height is a central goal of atlas construction. Traditional atlases operate on 2D or 3D images and model dense deformation fields to establish spatial correspondence and capture population variability (Joshi et al., 2004; Rakic et al., 2025). In many settings, however, once correspondence is established, local geometry can be summarized as a scalar quantity — such as cortical thickness in the brain (Fischl & Dale, 2000) or cross-sectional area in tubular organs Hong et al. (2013). This reduces atlas construction to modeling how a scalar varies across spatial location and population covariates.

Tubular anatomies — airways, blood vessels, intestines — are a natural setting for this formulation. Correspondence is established by extracting centerlines (Fig. 1a), and geometry at each depth is summarized by cross-sectional area (CSA) (Saboo et al., 2009; Hong et al., 2013; Grélard et al., 2017) (Fig. 1b), yielding a scalar function of a single

(a)    (b)

Figure 1: **Pediatric airways.** (a) A 3D airway surface with its extracted centerline; anatomical landmarks are labeled at their normalized depth $x \in [0, 1]$ from choana to carina. (b) Cross-sectional area profiles plotted against normalized depth for individual airways (gray curves), with dashed lines indicating landmark positions.

---

*Correspondence: `jyn@cs.unc.edu`

spatial coordinate. This enables clinically important comparisons — for example, assessing CSA at the subglottis across hundreds of pediatric airways to detect subglottic stenosis (Hong et al., 2013). Yet it remains challenging to build a unified representation that captures how these scalars vary jointly across space and covariates while quantifying uncertainty.

Building such an atlas requires addressing several interrelated challenges:

**Spatial-level: Do population trends and variance vary by location?** Different anatomical locations may exhibit distinct population trends and covariate effects. For instance, the relationship between age and CSA may differ between the subglottis and carina. The model must capture such spatial dependence rather than treating each location independently. *`LucidAtlas` captures spatial dependence, yielding location-specific trends and variance.*

**Covariate-level: Which covariates influence the anatomy, and how?** Isolating individual covariate effects (e.g., age versus sex) is frequently a goal of medical studies, requiring representations that are inherently interpretable and transparent by construction (Rudin, 2019). Moreover, in practice users often seek to understand how anatomy changes with respect to *a subset of covariates* while marginalizing over others. When covariates are statistically dependent, ignoring this dependence may lead to misleading conclusions; covariate interpretations must therefore be *dependence-aware. `LucidAtlas` is interpretable by design, separating covariate effects and explicitly accounting for covariate dependence.*

**Population-level: How does the population distribution vary with covariates?** Not all population variation is attributable to observed covariates. The residual variability may itself be heteroscedastic — differing across covariate values and anatomical locations. For example, population variance may differ with age and airway location when constructing a pediatric airway atlas. *`LucidAtlas` models the population distribution conditioned on covariates, yielding a covariate-conditioned mean and spatially varying uncertainty.*

**Subject-level: Can the atlas enable patient-specific analysis?** An atlas with reliable uncertainty estimates opens the door to individualized applications. Can the model predict anatomy at time $T_1$ given an observation at $T_0$? Can it flag abnormal cases as out-of-distribution? Both tasks require covariate-conditioned heteroscedastic uncertainty. *`LucidAtlas` uses covariate-conditioned means and uncertainties to enable per-subject forecasting and OOD detection.*

We present `LucidAtlas`, a unified framework that simultaneously addresses all four levels for atlas construction over scalar anatomical measurements. The main contributions are as follows:

(1) We unify spatial awareness, covariate interpretability, and heteroscedastic uncertainty within a single atlas representation, creating a versatile tool for modeling how scalar quantities vary across space and population covariates. (2) We identify the risks of using Neural Additive Models (NAMs) when covariates are dependent and propose a dependence-aware marginalization approach that supports conditioning on arbitrary covariate subsets. (3) We enable downstream clinical applications including individualized temporal prediction and spatially aware out-of-distribution detection. (4) We validate on one synthetic dataset and two real-world medical datasets — a Pediatric Airway Shape dataset and the OASIS Brain Volume dataset (Marcus et al., 2007) — demonstrating superior performance over baselines.

## 2 Related Work

We first introduce the three most related research directions to our `LucidAtlas` approach.

**Additive Models.** Model-agnostic methods, such as Partial Dependence (Friedman, 2001), SHAP (Lundberg, 2017), and LIME (Ribeiro et al., 2016), offer a standardized approach to explaining machine learning predictions. However, when applied to deep neural networks (DNNs), these methods may fail to provide faithful representations of the full complexity of DNNs (Rudin, 2019). A more structurally interpretable alternative involves leveraging Generalized Additive Models (GAMs) (Hastie, 2017), where a response variable

| Method | Covariate- Dependence-Aware | Subject- Ind. Pred. | OOD Det. | Population- Hetero.+Aleatoric | Spatial- Spa. Dep. |
|---|---|---|---|---|---|
| NAM | ✗ | ✗ | ✗ | ✗ | ✗ |
| LA-NAM (Bouchiat et al., 2023) | ✗ | ✓̇ | ✗ | ✗ | ✗ |
| NAMLSS (Thielmann et al., 2024) | ✗ | ✓̇ | ✓̇ | ✓ | ✗ |
| NAISR (Jiao et al., 2023) | ✗ | ✗ | ✗ | ✗ | ✓ |
| LucidAtlas(Ours) | ✓ | ✓ | ✓ | ✓ | ✓ |

Table 1: Comparison of interpretable representations based on the desirable properties discussed in Sec.1. **Ind. Pred.** indicates individualized prediction, i.e., whether the model can predict a response for time $T_1$ given an earlier observation at time $T_0$. **OOD Det.** indicates whether the model is capable of out-of-distribution detection. **Hetero.+Aleatoric** indicates whether the model considers heteroscedasticity when modeling aleatoric uncertainty. **Spa. Dep.** indicates spatial dependence. A ✓ indicates that a model has a property; a ✗ indicates that it does not. A ✓̇ indicates that the model potentially possesses the property, but it was not explored in the original study. Only LucidAtlas possesses and explores all the desired properties.

$y$ is modeled using an additive structure:

$$E[y \mid c_1, ..., c_N] = h(\beta_0 + f_1(c_1) + \cdots + f_N(c_N)).$$ (1)

Here, $h(\cdot)$ is the inverse of the link function (a form of activation function); $\beta_0$ denotes the intercept and $f_i(\cdot)$ represent independent functions for the $i^{th}$ covariate. Neural Additive Models (NAMs) (Agarwal et al., 2020; Jiao et al., 2023) build upon this framework, offering enhanced interpretability while maintaining the flexibility of neural networks. Specifically, for NAMs the functions $f_i(\cdot)$ are deep neural networks. NAISR (Jiao et al., 2023) pioneers the use of NAMs to capture spatial deformations with respect to an estimated atlas shape that is modulated by covariates. *LucidAtlas extends this concept by integrating NAMs to construct an atlas that captures population trends and uncertainties with spatial dependence.*

**Uncertainty and Heteroscedasticity.** Two types of uncertainties need to be distinguished: epistemic uncertainty captures model uncertainty, whereas aleatoric uncertainty captures uncertainty in the data (Hüllermeier & Waegeman, 2021). Existing interpretable models focus primarily on epistemic uncertainty, e.g., via ensembling (Agarwal et al., 2020) or Laplace approximation (Bouchiat et al., 2023). Moreover, most assume homoscedasticity (Wooldridge et al., 2016), i.e., constant variance across covariates and spatial locations. In atlas construction, however, aleatoric uncertainty is heteroscedastic—for example, population variance in airway cross-sectional area increases with age. NAMLSS (Thielmann et al., 2024) is, to our knowledge, the only additive model that considers heteroscedastic aleatoric uncertainty, using NAMs to approximate the parameters $\{\theta^{(n)}\}$ of a chosen data distribution as $\theta^{(n)} = h^{(n)}(\beta^{(n)} + \sum_{i=1}^{N} f_i^{(n)}(c_i))$, where $\theta^{(n)}$ can be the mean and variance of a Gaussian distribution. *LucidAtlas incorporates explicit spatial dependence into NAMLSS and introduces a marginalization approach that enables correct covariate interpretation in the presence of covariate dependence, without relying on the independence assumption underlying standard NAM interpretations.*

**Spatial Dependence Modeling.** Modeling spatial dependence in regression is well studied outside the additive model literature. A common approach is to introduce a spatial random effect — for example, as a Gaussian process for continuous domains (Rasmussen, 2003) or a conditional autoregressive (CAR) model for discrete areal data (Banerjee et al., 2003) — as an additive term that captures spatial correlations independently of covariates. While effective at modeling spatial structure, such formulations treat covariate effects as global fixed effects that do not vary across spatial locations. *LucidAtlas instead lets each covariate subnetwork $f_i(c_i, x)$ take the spatial coordinate as input, so that covariate effects are simultaneously nonlinear, per-covariate disentangled, and spatially varying.*

Tab. 1 compares LucidAtlas to related interpretable models with respect to the discussed properties above. A more comprehensive discussion of related work is available in Sec. S.1 of the Supplementary Material.

## 3   Method

### 3.1   Problem Formulation

We consider a set of anatomical functions $\mathcal{Y}^k(x)$ as our data samples, each mapping a spatial input $x$ to an observed value $y$. In this work, we focus primarily on the setting where $\mathcal{Y}^k$ maps a one-dimensional spatial domain to a one-dimensional output, i.e., $\mathcal{Y}^k(x) : \mathbb{R} \to \mathbb{R}$. Each subject $k$ is also associated with a covariate vector $\boldsymbol{c}^k = [c_1^k, \dots, c_N^k]$, representing relevant attributes such as age, weight, or other clinical variables.

Our goal is to construct an atlas representation that addresses the questions outlined in Section 1 by learning the mapping from covariates $\boldsymbol{c}$ and spatial location $x$ to observations $y$, while simultaneously accounting for heteroscedastic uncertainties within the population, represented as a normal distribution $p(y \mid \boldsymbol{c}, x) = \mathcal{N}(f^m(\boldsymbol{c}, x), f^v(\boldsymbol{c}, x))$, where $f^m(\boldsymbol{c}, x)$ and $f^v(\boldsymbol{c}, x)$ are the spatially dependent mean and variance for a given covariate $\boldsymbol{c}$.

In addition to modeling the full conditional distribution $p(y \mid \boldsymbol{c}, x)$, we aim to understand how individual covariates, and subsets of covariates, influence the observed value $y$ at each spatial location. To this end, we examine conditional distributions such as $p(y \mid c_i, x)$ and $p(y \mid \boldsymbol{c}_{\mathcal{S}}, x)$, where $\boldsymbol{c}_{\mathcal{S}}$ denotes an arbitrary subset of covariates. We refer to these quantities as *Marginalized Covariate Effects* [1].

### 3.2   `LucidAtlas` Formulation

**Roadmap.**   We develop `LucidAtlas` based on NAMLSS (Thielmann et al., 2024), and introduce several key extensions to ① introduce spatial awareness (Sec. 3.2.1), ② support quantification of marginalized covariate effects for arbitrary subsets of covariates (Secs. 3.3 - 3.4), and ③ enable downstream applications such as individualized temporal prediction (Sec. 3.5.3) and spatially aware out-of-distribution detection (Sec. 3.5.2). Together, these extensions make `LucidAtlas` a practical tool for clinically relevant tasks where interpretable, uncertainty-aware modeling is critical.

#### 3.2.1   Introducing Spatial Dependence

This section addresses spatial dependence, which is not explicitly modeled in NAMLSS (Thielmann et al., 2024). To achieve this, we introduce neural subnets $\{f_i(c_i, x)\}$ that predict the distributional parameters of $p(y \mid \boldsymbol{c}, x)$. Each subnetwork $f_i(c_i, x)$ has two outputs: $f_i^m(c_i, x)$ and $f_i^v(c_i, x)$, which capture the contribution from $c_i$ at location $x$ to the mean and variance of $p(y \mid \boldsymbol{c}, x)$ respectively. The overall population mean and variance are then obtained by summing these individual contributions as

$$f^m(\boldsymbol{c}, x) = \sum_{i=1}^{N} f_i^m(c_i, x) + b^m(x), \quad f^v(\boldsymbol{c}, x) = \sum_{i=1}^{N} f_i^v(c_i, x) + b^v(x), \tag{2}$$

where $b^m(x)$ and $b^v(x)$ represent the bias terms. By explicitly modeling spatial dependence, `LucidAtlas` extends NAMLSS to spatial atlas construction.

**Loss Function.**   We optimize the subnetworks $\{f_i\}_{i=1}^{N}$ by minimizing a total loss that combines the negative log-likelihood (NLL) of the predicted Gaussian distribution with regularization on the outputs of individual subnetworks.

The NLL term of the overall Gaussian is defined as

$$\mathcal{L}_{\text{NLL}} = \frac{1}{|\mathcal{D}|} \sum_{(\boldsymbol{c}, x, y) \in \mathcal{D}} \left[ \frac{1}{2} \log(2\pi \cdot f^v(\boldsymbol{c}, x)) + \frac{(y - f^m(\boldsymbol{c}, x))^2}{2 \cdot f^v(\boldsymbol{c}, x)} \right], \tag{3}$$

where $y$ is the observation at location $x$ given the covariates.

---

[1] Rather than training a different model for each possible subset, which would be both computationally intensive, we employ a unified modeling approach that supports subset-wise marginalization directly. This strategy improves efficiency and ensures consistency across different conditional views.

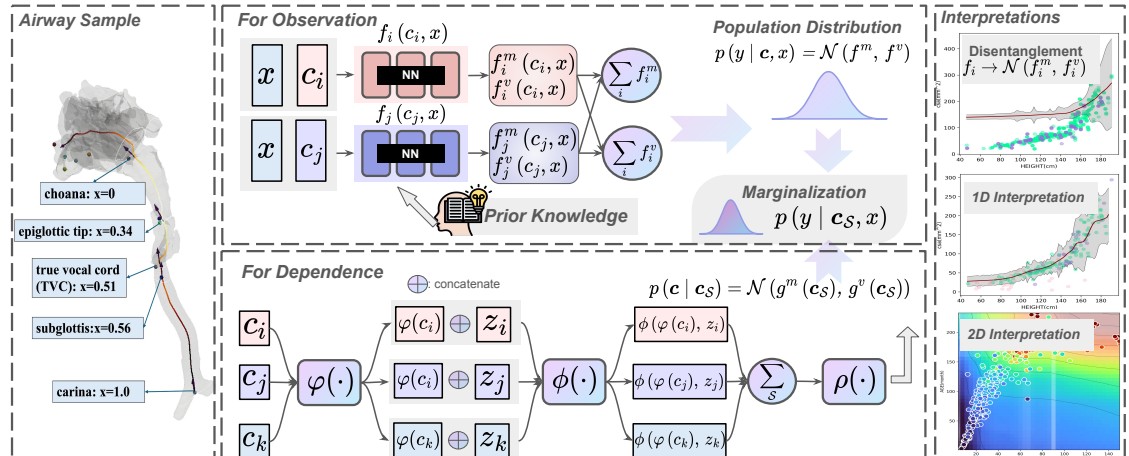

Figure 2: `LucidAtlas`: Learning an Uncertainty-Aware, Covariate-Disentangled, Individualized Atlas Representation. ① As an example use case, we depict an airway with its anatomical landmarks at different depths (i.e., anatomical locations) along its centerline (Hong et al., 2013). ② During training, each subnetwork $f_i(c_i, x)$ receives the location $x$ and covariate $c_i$ as input to predict the covariate-specific distributional parameters $f_i^m$ and $f_i^v$ (mean and variance), which are added to obtain the overall distributional parameters to capture the population trend and variation as $f^m = \sum_i f_i^m$ and $f^v = \sum_i f_i^v$ respectively. ③ The goal of marginalization is to obtain distributions conditioned on desired covariates $p(y \mid \boldsymbol{c}_\mathcal{S}, x)$, by integrating out the potentially dependent covariates $\{c_i\}_{i \notin \mathcal{S}}$. A DeepSet-based parameterization (Zaheer et al., 2018) $g(\boldsymbol{c}_\mathcal{S})$ is used to model the covariate dependence as $p(\boldsymbol{c} \mid \boldsymbol{c}_\mathcal{S})$ given an arbitrary subset of covariates $\boldsymbol{c}_\mathcal{S}$. ④ `LucidAtlas` supports: (i) covariate disentanglement via additive components $\{f_i\}$; (ii) a marginalized covariate effect conditioned on a single covariate (e.g., height on CSA); (iii) a marginalized effect conditioned on an arbitrary set of covariates. ⑤ When domain knowledge implies a monotonic influence, we use monotonic neural networks; otherwise, MLP subnetworks.

The regularization terms are

$$\mathcal{L}_{\text{feat}}^m = \frac{1}{|\mathcal{D}|} \sum_{(c_i, x) \in \mathcal{D}} \frac{1}{N} \sum_{i=1}^N f_i^m(c_i, x)^2, \quad \mathcal{L}_{\text{feat}}^v = \frac{1}{|\mathcal{D}|} \sum_{(c_i, x) \in \mathcal{D}} \frac{1}{N} \sum_{i=1}^N f_i^v(c_i, x)^2. \tag{4}$$

These regularization terms help mitigate identifiability issues by discouraging redundant subnetwork outputs, encouraging a stable and separated representation of the covariate effect (Agarwal et al., 2020).

The total loss is then

$$\mathcal{L}_{\text{total}} = \mathcal{L}_{\text{NLL}} + \lambda(\mathcal{L}_{\text{feat}}^m + \mathcal{L}_{\text{feat}}^v), \lambda > 0. \tag{5}$$

### 3.2.2 Prior Knowledge

In clinical settings where domain knowledge suggests monotonic relationships (e.g., airway cross-sectional area increasing with age and height), this prior can be incorporated via the modeling ansatz. Assuming that the response $y$ has a monotonically increasing relationship with respect to a particular covariate $c_j$ (while keeping all other covariates fixed), monotonicity prior can be incorporated via the modeling ansatz

$$\frac{\partial f^m(\boldsymbol{c}, x)}{\partial c_j} = \sum_{i=1}^N \frac{\partial f_i^m(c_i, x)}{\partial c_j} = \frac{\partial f_j^m(c_j, x)}{\partial c_j} \geq 0. \tag{6}$$

As illustrated in Fig. 2, $f_j(c_j)$ can be parameterized using a monotonic Lipschitz neural network (Kitouni et al., 2023) when such prior monotonicity information is available. This design guarantees the monotonicity of $f_j(c_j)$ by construction and ensures that the interpretations derived from the NAMs align with human prior knowledge.

### 3.3 Pitfalls of NAM Interpretations

The underlying assumption behind NAMs is that each covariate contributes independently to the outcome. In most real-world applications, such as our airway atlas construction problem described in Sec. 3.1, the independence between covariates cannot be guaranteed. *A natural question is whether accepting the independence assumption unquestioningly and directly using NAMs is appropriate.*

We can investigate this problem with a toy example: $y = sin(c_1) + c_2 + \epsilon$ where $\epsilon$ is a noise term and $c_1$, $c_2$ are covariates that influence the observed outcome $y$. Assuming there is a NAM that already fits this function well, the subnetworks capture $f_1(c_1) = \sin(c_1)$ and $f_2(c_2) = c_2$ and thus approximate $y$ with $f(c_1, c_2) = f_1(c_1) + f_2(c_2)$.

If we want to interpret the population trend of $y$ only with respect to $c_1$, we need to marginalize $c_2$ out as,

$$F_1(c_1) = \int [f_1(c_1) + f_2(c_2)] p(c_2 \mid c_1) \, \mathrm{d}c_2 = \underbrace{f_1(c_1)}_{\text{Interpretation from NAMs}} + \underbrace{\int f_2(c_2) p(c_2 \mid c_1) \, \mathrm{d}c_2}_{\text{Interpretation from Dependence: } := h_1(c_1)} \tag{7}$$

where $h_1(c_1)$ measures how the dependence between $c_1$ and $c_2$ influences the marginalization $F_1(c_1)$. We can see from Eq. (7) that $F_1(c_1)$ is composed of the interpretation from the NAM's subnetwork plus the interpretation from the dependence between $c_1$ and $c_2$ as $h_1(c_1)$.

**If $c_1$ and $c_2$ are *independent*,** $h_1(c_1) = \int f_2(c_2) p(c_2 \mid c_1) \, \mathrm{d}c_2 = \int f_2(c_2) p(c_2) \, \mathrm{d}c_2 = constant$, which means the marginalization is the actual covariate disentanglement in Eq. (2) plus a constant. **If $c_1$ and $c_2$ are *dependent*,** $h_1(c_1)$ is a function of $c_1$ which no longer needs to be a constant and could be a non-trivial function of $c_1$ arising from the inherent dependence between $c_1$ and $c_2$. Therefore, considering the relationship between $c_1$ and $c_2$ is crucial when using either covariate by itself to interpret the population trend.

In summary, the structurally separated covariate effects of NAMs, combined with those effects contributed by covariate dependence, shape human-understandable explanations that align with population distributions. *While ignoring the potential dependence in NAMs may not impact prediction performance, it can result in ambiguous or misleading interpretations when analyzing population trends.* More analysis is available in Sec. S.2.2 in the Supplementary Material.

### 3.4 Marginalization Approach

#### 3.4.1 Marginalized Covariate Effects

The section introduces our marginalization approach, which enables covariate analysis conditioning on any arbitrary subset of covariates. The observation $y$ can be formulated as

$$y = f^m(\boldsymbol{c}, x) + f^v(\boldsymbol{c}, x) \cdot \epsilon, \ \epsilon \sim \mathcal{N}(0, 1). \tag{8}$$

We expand the two-covariate case in Sec. 3.3 to the multi-covariate setting. Let $\mathcal{S} \subseteq \{1, \ldots, N\}$ denote a set of covariate indices. We define $\boldsymbol{c}_{\mathcal{S}} = (c_i)_{i \in \mathcal{S}}$ as the subvector of $\boldsymbol{c} = (c_1, \ldots, c_N)$ containing the covariates indexed by $\mathcal{S}$. Thus, $\mathcal{S}$ specifies which covariates are selected, while $\boldsymbol{c}_{\mathcal{S}}$ denotes their corresponding values as an ordered vector. Note that while $\mathcal{S}$ is a set, the notation $\boldsymbol{c}_{\mathcal{S}}$ explicitly preserves the indexing and order of the selected covariates. For simplicity, we denote the complement by $\boldsymbol{c}_{-\mathcal{S}} = (c_i)_{i \notin \mathcal{S}}$. We aim to derive the mean and variance of the conditional distribution $p(y \mid \boldsymbol{c}_{\mathcal{S}}, x)$, as $f^m(\boldsymbol{c}_{\mathcal{S}}, x)$ and $f^v(\boldsymbol{c}_{\mathcal{S}}, x)$ respectively. The **full** derivation is given in Sec. S.2.3.

**Mean of $p(y \mid \boldsymbol{c}_{\mathcal{S}}, x)$.** We now derive the conditional mean of $p(y \mid \boldsymbol{c}_{\mathcal{S}}, x)$ by marginalizing over the complementary covariates $\boldsymbol{c}_{-\mathcal{S}}$. For consistency with our earlier notation, we denote this conditional mean by

$f^m(\boldsymbol{c}_\mathcal{S}, x)$. It is derived as

$$f^m(\boldsymbol{c}_\mathcal{S}, x) = \mathbb{E}_{\boldsymbol{c}_{-\mathcal{S}}|\boldsymbol{c}_\mathcal{S}}[f^m(\boldsymbol{c}, x) + f^v(\boldsymbol{c}, x) \cdot \boldsymbol{\epsilon}] = \mathbb{E}_{\boldsymbol{c}_{-\mathcal{S}}|\boldsymbol{c}_\mathcal{S}}[f^m(\boldsymbol{c}, x)] = \int f^m(\boldsymbol{c}, x) p(\boldsymbol{c}_{-\mathcal{S}} \mid \boldsymbol{c}_\mathcal{S}) \, \mathrm{d}\boldsymbol{c}_{-\mathcal{S}}$$

$$= \int (\sum_{i=1}^N f_i^m(c_i, x)) p(\boldsymbol{c}_{-\mathcal{S}} \mid \boldsymbol{c}_\mathcal{S}) \, \mathrm{d}\boldsymbol{c}_{-\mathcal{S}} + b^m(x) = \sum_{i\in\mathcal{S}} f_i^m(c_i, x) + \underbrace{\int (\sum_{i\notin\mathcal{S}} f_i^m(c_i, x)) p(\boldsymbol{c}_{-\mathcal{S}} \mid \boldsymbol{c}_\mathcal{S}) \, \mathrm{d}\boldsymbol{c}_{-\mathcal{S}}}_{:=H} + b^m(x), \quad (9)$$

where $f_i^m(c_i, x)$ represents the interpretation from the additive subnetwork $f_i$ of `LucidAtlas`, while $H$ accounts for the contributions from the dependence between the covariates which can be further simplified as follows

$$H = \sum_{i\notin\mathcal{S}} \int f_i^m(c_i, x) \left( \int p(\boldsymbol{c}_{-\mathcal{S}\backslash\{i\}}, c_i \mid \boldsymbol{c}_\mathcal{S}) \, dc_{-\mathcal{S}\backslash\{i\}} \right) dc_i = \sum_{i\notin\mathcal{S}} \int f_i^m(c_i, x) \, p(c_i \mid \boldsymbol{c}_\mathcal{S}) \, dc_i \,. \quad (10)$$

In the second step, we marginalize all covariates in $\boldsymbol{c}_{-\mathcal{S}}$ except for $c_i$ (denoted as $\boldsymbol{c}_{-\mathcal{S}\backslash\{i\}}$). Eq. (10) indicates that even when multiple covariates are involved, only the univariate conditional distributions $p(c_i \mid \boldsymbol{c}_\mathcal{S})$ are required to compute $f^m(\boldsymbol{c}_\mathcal{S}, x)$ as a consequence of the additive design of a NAM, which simplifies computations (discussed in Sec. 3.4.2). Therefore,

$$f^m(\boldsymbol{c}_\mathcal{S}, x) = \sum_{i\in\mathcal{S}} f_i^m(c_i, x) + \sum_{i\notin\mathcal{S}} \int f_i^m(c_i, x) p(c_i \mid \boldsymbol{c}_\mathcal{S}) \, \mathrm{d}c_i + b^m(x) \,. \quad (11)$$

**Variance of $p(y \mid \boldsymbol{c}_\mathcal{S}, x)$.** Now, we derive the conditional variance of $p(y \mid \boldsymbol{c}_\mathcal{S}, x)$. The *law of total variance* is $\mathrm{Var}(Y) = \mathbb{E}[\mathrm{Var}(Y \mid X)] + \mathrm{Var}(\mathbb{E}[Y \mid X])$, which states that the total variance of a random variable $Y$ can be broken into two parts: ① the **expected variance of $Y$ given $X$**, which represents how much $Y$ fluctuates around its mean for each specific value of $X$; and ② **the variance of the expected value of $Y$ given $X$**, which measures how much the conditional mean itself varies as $X$ changes.

In our case, we apply this to the predictive distribution $p(y \mid \boldsymbol{c}_\mathcal{S}, x)$, where only a subset $\boldsymbol{c}_\mathcal{S}$ of covariates is known, and the rest of the covariates $\boldsymbol{c}_{-\mathcal{S}}$ are marginalized. Conditioning on $\boldsymbol{c}_\mathcal{S}$, we write:

$$f^v(\boldsymbol{c}_\mathcal{S}, x) = \mathrm{Var}(y \mid \boldsymbol{c}_\mathcal{S}, x) = \underbrace{\mathbb{E}_{\boldsymbol{c}_{-\mathcal{S}}|\boldsymbol{c}_\mathcal{S}} [\mathrm{Var}(y \mid \boldsymbol{c}_{-\mathcal{S}}, \boldsymbol{c}_\mathcal{S}, x)]}_{①:=\tilde{\sigma}_E^2(\boldsymbol{c}_\mathcal{S}, x)} + \underbrace{\mathrm{Var}_{\boldsymbol{c}_{-\mathcal{S}}|\boldsymbol{c}_\mathcal{S}} (\mathbb{E}[y \mid \boldsymbol{c}_{-\mathcal{S}}, \boldsymbol{c}_\mathcal{S}, x])}_{②:=\tilde{\sigma}_V^2(\boldsymbol{c}_\mathcal{S}, x)} \,. \quad (12)$$

Using the additive structure $f^v(\boldsymbol{c}, x) = \sum_{i=1}^N f_i^v(c_i, x) + b^v(x)$, we compute ① as:

$$\tilde{\sigma}_E^2(\boldsymbol{c}_\mathcal{S}, x) = \mathbb{E}_{\boldsymbol{c}_{-\mathcal{S}}|\boldsymbol{c}_\mathcal{S}} [f^v(\boldsymbol{c}, x)] = \int f^v(\boldsymbol{c}, x) p(\boldsymbol{c}_{-\mathcal{S}} \mid \boldsymbol{c}_\mathcal{S}) d\boldsymbol{c}_{-\mathcal{S}} = \sum_{i\in\mathcal{S}} f_i^v(c_i, x) + \sum_{i\notin\mathcal{S}} \int f_i^v(c_i, x) \, p(c_i \mid \boldsymbol{c}_\mathcal{S}) \, \mathrm{d}c_i + b^v(x). \quad (13)$$

We now explain the second term, $\tilde{\sigma}_V^2(\boldsymbol{c}_\mathcal{S}, x)$, which corresponds to the variance of the conditional mean function $\mathbb{E}[y \mid \boldsymbol{c}_{-\mathcal{S}}, \boldsymbol{c}_\mathcal{S}, x] = f^m(\boldsymbol{c}, x)$. Because we are conditioning on $\boldsymbol{c}_\mathcal{S}$, the functions $f_i^m(c_i, x)$ for $i \in \mathcal{S}$ are deterministic, so only the terms for $i \notin \mathcal{S}$ contribute to variance. Therefore:

$$\tilde{\sigma}_V^2(\boldsymbol{c}_\mathcal{S}, x) = \mathrm{Var}_{\boldsymbol{c}_{-\mathcal{S}}|\boldsymbol{c}_\mathcal{S}} \left( \sum_{i\notin\mathcal{S}} f_i^m(c_i, x) \right) = \sum_{i\notin\mathcal{S}} \underbrace{\mathrm{Var}_{c_i|\boldsymbol{c}_\mathcal{S}} (f_i^m(c_i, x))}_{③} + \sum_{\substack{j_1\neq j_2 \\ j_1,j_2\notin\mathcal{S}}} \underbrace{\mathrm{Cov}_{(c_{j_1}, c_{j_2})|\boldsymbol{c}_\mathcal{S}} (f_{j_1}^m(c_{j_1}, x), f_{j_2}^m(c_{j_2}, x))}_{④}, \quad (14)$$

where

$$③ = \int (f_i^m(c_i, x) - \tilde{\mu}_i(\boldsymbol{c}_\mathcal{S}, x))^2 p(c_i \mid \boldsymbol{c}_\mathcal{S}) \, \mathrm{d}c_i \,, \quad \tilde{\mu}_i(\boldsymbol{c}_\mathcal{S}, x) = \int f_i^m(c_i, x) p(c_i \mid \boldsymbol{c}_\mathcal{S}) dc_i \,, \quad (15)$$

$$④ = \iint f_{j_1}^m(c_{j_1}, x) f_{j_2}^m(c_{j_2}, x) \, p(c_{j_1}, c_{j_2} \mid \boldsymbol{c}_\mathcal{S}) \, \mathrm{d}c_{j_1} \, \mathrm{d}c_{j_2} - \tilde{\mu}_{j_1}(\boldsymbol{c}_\mathcal{S}, x) \, \tilde{\mu}_{j_2}(\boldsymbol{c}_\mathcal{S}, x) \,. \quad (16)$$

From Eqs. (13) to (16), all integrals are performed over $c_i$ or $(c_{j_1}, c_{j_2})$, conditioned on $\boldsymbol{c}_{\mathcal{S}}$, and are evaluated under the corresponding multivariate distributions. Eqs. (13) to (16) reveal that, rather than sampling from the full covariate space, it suffices to sample from conditional distributions such as $p(c_i \mid \boldsymbol{c}_{\mathcal{S}})$ and $p(c_{j_1}, c_{j_2} \mid \boldsymbol{c}_{\mathcal{S}})$ in order to compute the marginalized predictive distribution $p(y \mid \boldsymbol{c}_{\mathcal{S}}, x)$. Our proposed approach allows marginalization over arbitrary subsets of covariates without requiring model retraining.

### 3.4.2 Covariate Dependence Modeling with DeepSets

The marginalized covariate effects derived in Sec. 3.4.1 require evaluating conditional distributions such as $p(c_i \mid c_{\mathcal{S}})$ and $p(c_{j_1}, c_{j_2} \mid c_{\mathcal{S}})$ for arbitrary subsets $S$. This calls for a flexible model $g(\boldsymbol{c}_{\mathcal{S}})$ that can model conditional distributions such as $p(c_i \mid c_{\mathcal{S}})$ and $p(c_{j_1}, c_{j_2} \mid c_{\mathcal{S}})$ with arbitrary subsets $\mathcal{S} \subseteq \{1, \ldots, N\}$ of observed covariates as input. We adopt a DeepSets-based parameterization (Zaheer et al., 2018) that works over sets and is therefore particularly well suited to condition based on arbitrary subsets of the covariates by allowing for variable length inputs.

Specifically, each observed covariate $c_i$ is represented with a trainable ID embedding $z_i$ (fixed for each covariate type) and a value embedding from a shared value encoder network $\varphi(c_i)$. Another neural network $\phi(\cdot)$ takes $z_i$ and $\varphi(c_i)$ to produce a unified representation $\phi(z_i, \varphi(c_i))$ for each covariate. These per-covariate representations are then summed and fed into a decoder network $\rho(\cdot)$, which predicts the mean and covariance of $p(\boldsymbol{c} \mid \boldsymbol{c}_{\mathcal{S}})$ as $(g^m(\boldsymbol{c}_{\mathcal{S}}), g^v(\boldsymbol{c}_{\mathcal{S}})) = \rho(\sum_{i \in \mathcal{S}} \phi(z_i, \varphi(c_i)))$. We assume $p(\boldsymbol{c} \mid \boldsymbol{c}_{\mathcal{S}})$ is a multivariate Gaussian distribution, i.e., $p(\boldsymbol{c} \mid \boldsymbol{c}_{\mathcal{S}}) = \mathcal{N}(g^m(\boldsymbol{c}_{\mathcal{S}}), g^v(\boldsymbol{c}_{\mathcal{S}}))$.

The conditional distribution $p(\boldsymbol{c}_{\mathcal{K}} \mid \boldsymbol{c}_{\mathcal{S}})$ is given by $p(\boldsymbol{c}_{\mathcal{K}} \mid \boldsymbol{c}_{\mathcal{S}}) = \mathcal{N}(g^m_{\mathcal{K}}(\boldsymbol{c}_{\mathcal{S}}), g^v_{\mathcal{K}\mathcal{K}}(\boldsymbol{c}_{\mathcal{S}}))$, where $g^m_{\mathcal{K}}(\boldsymbol{c}_{\mathcal{S}})$ denotes the subvector of the mean $g^m(\boldsymbol{c}_{\mathcal{S}})$ corresponding to indices $\mathcal{K}$, and $g^v_{\mathcal{K}\mathcal{K}}(\boldsymbol{c}_{\mathcal{S}})$ is the corresponding covariance submatrix. These are obtained from the full mean $g^m(\boldsymbol{c}_{\mathcal{S}})$ and covariance $g^v(\boldsymbol{c}_{\mathcal{S}})$ using the conditional distribution formula of multivariate Gaussians (Anderson, 2003).

The networks are optimized via the negative log-likelihood (NLL) of the observed covariate data under the predicted Gaussian:

$$\mathcal{L}_{\text{cov}} = \frac{1}{2}\left[N \log(2\pi) + \log \det(g^v(\boldsymbol{c}_{\mathcal{S}})) + \left(\boldsymbol{c} - g^m(\boldsymbol{c}_{\mathcal{S}})\right)^\top \left(g^v(\boldsymbol{c}_{\mathcal{S}})\right)^{-1} \left(\boldsymbol{c} - g^m(\boldsymbol{c}_{\mathcal{S}})\right)\right], \tag{17}$$

where $N$ is the dimensionality of $\boldsymbol{c}$.

The model generalizes to partially observed test-time inputs without requiring explicit imputation. This design leverages DeepSets not only for its invariance to covariate order, but more crucially for its ability to encode covariate sets of variable size in a principled and differentiable way.

**Approximation.** The integrals for $f^m(\boldsymbol{c}_{\mathcal{S}}, x)$ (see Eq. (11)), $\tilde{\sigma}^2_E(\boldsymbol{c}_{\mathcal{S}}, x)$ (see Eq. (13)) and $\tilde{\sigma}^2_V(\boldsymbol{c}_{\mathcal{S}}, x)$ (see Eqs. (14) to (16)) can be approximated using Monte Carlo sampling. E.g. for $f^m(\boldsymbol{c}_{\mathcal{S}}, x)$, for each covariate $c_i$ one can sample $L$ values $\{\hat{c}^l_i\}_{l=1,\ldots,L}$ from the distribution of covariates $p(c_i \mid \boldsymbol{c}_{\mathcal{S}})$ given by $g(\boldsymbol{c}_{\mathcal{S}})$ to approximate $f^m(\boldsymbol{c}_{\mathcal{S}}, x)$ as $f^m(\boldsymbol{c}_{\mathcal{S}}, x) \approx \sum_{i \in \mathcal{S}} f^m_i(c_i, x) + \frac{1}{L}\sum_{i \notin \mathcal{S}}\sum_{l=1}^{L} f^m_i(\hat{c}^l_i, x) + b^m(x)$.

**Computational Complexity.** Assume we have $N$ covariates and $L$ samples for each covariate dimension. Marginalizing covariate effect using a neural additive model (NAM) requires $\mathcal{O}(LN)$ model queries to approximate the mean and $\mathcal{O}(L^2 N^2)$ to approximate the variance, which is computationally efficient. In contrast, a black-box model without structural assumptions would require evaluating all possible combinations of covariate configurations, resulting in a complexity of $\mathcal{O}(L^N)$—infeasible for even moderately large $N$.

As a result, `LucidAtlas` can provide the conditional mean $f^m(\boldsymbol{c}_{\mathcal{S}}, x)$ and total predictive variance $f^v(\boldsymbol{c}_{\mathcal{S}}, x) = \tilde{\sigma}^2_E + \tilde{\sigma}^2_V$ for any desired covariate subset $\mathcal{S}$, allowing us to parameterize the conditional predictive distribution $p(y \mid \boldsymbol{c}_{\mathcal{S}}, x)$. `LucidAtlas` extends classical NAM interpretations, retains uncertainty estimates, and enables efficient post-hoc analysis of covariate influence for any single covariate or covariate subset.

### 3.5 Additional Applications of `LucidAtlas`

#### 3.5.1 Imputation

Our framework naturally supports covariate imputation by querying the conditional distribution $p(\boldsymbol{c} \mid \boldsymbol{c}_{\mathcal{S}})$ using the DeepSets-based covariate network described in Sec. 3.4.2. For any missing covariate $c_i$, we compute the conditional mean vector $g^m(\boldsymbol{c}_{\mathcal{S}})$ given the observed subset $\boldsymbol{c}_{\mathcal{S}}$, and impute $c_i$ using $g_k^m(\boldsymbol{c}_{\mathcal{S}})$, the $k$-th element of the mean vector.

#### 3.5.2 Out-of-Distribution Detection

A natural feature of `LucidAtlas` is its ability to perform **spatially aware out-of-distribution (OOD) detection**. Given a spatial location $x$ and subject-level covariates $\boldsymbol{c}$, we define the population-level distribution via the predicted mean $f^{\mathrm{m}}(\boldsymbol{c}, x)$ and variance $f^{\mathrm{v}}(\boldsymbol{c}, x)$. For an anatomy $y(x)$, we compute the z-score at each location as: $z(\boldsymbol{c}, x; y) = \frac{y(x) - f^{\mathrm{m}}(\boldsymbol{c}, x)}{\sqrt{f^{\mathrm{v}}(\boldsymbol{c}, x)}}$. To detect localized model-data discrepancies, we define a subject-level OOD score by aggregating the z-score range across relevant spatial positions:

$$\Delta_{\mathrm{OOD}}(\boldsymbol{c}, y) = \min_x z(\boldsymbol{c}, x; y) - \max_x z(\boldsymbol{c}, x; y). \tag{18}$$

In the context of airway analysis, a strongly negative $\Delta_{\mathrm{OOD}}$ indicates that some regions are significantly narrower than expected — relative to model uncertainty — suggesting potential obstruction or anatomical mismatch with the training distribution. Ranking subjects by $\Delta_{\mathrm{OOD}}$ enables interpretable, uncertainty-aware identification of anatomically atypical cases.

#### 3.5.3 Individualized Prediction

One challenge in the context of atlas construction is to make individualized predictions when observations are predominantly limited to a single time point, i.e., when the atlas is built from cross-sectional data. We provide two alternative strategies for individualized prediction based on `LucidAtlas`:

**(1) Percentile-Preserving Prediction.** This strategy assumes that an individual's percentile rank in the conditional distribution remains unchanged across two adjacent time points. Let $\mathrm{F}(y \mid \boldsymbol{c}, x)$ denote the cumulative distribution function (CDF) of the population response $y$, conditioned on covariates $\boldsymbol{c}$ and spatial variable $x$. Then, for an individual observed at time $t$ with response $y^t$, we assume $\mathrm{F}(y^t \mid \boldsymbol{c}^t, x) = \mathrm{F}(y^{t+1} \mid \boldsymbol{c}^{t+1}, x)$. This is a plausible assumption in biological or anatomical growth, where relative status among peers tends to persist over short durations.

Given that F is defined as the CDF of a Gaussian distribution parameterized by mean $f^m(\boldsymbol{c}, x)$ and variance $f^v(\boldsymbol{c}, x)$, we can invert the CDF matching condition to obtain a predictive formula for $y^{t+1}$: $y^{t+1} \approx f^m(\boldsymbol{c}^{t+1}, x) + \sqrt{\frac{f^v(\boldsymbol{c}^{t+1}, x)}{f^v(\boldsymbol{c}^t, x)}} \cdot (y^t - f^m(\boldsymbol{c}^t, x))$.

**(2) Mean-Shift Prediction.** Alternatively, we assume the individual's deviation from the population mean remains constant over time. This leads to a simpler approximation: $y^{t+1} \approx y^t + \left(f^m(\boldsymbol{c}^{t+1}, x) - f^m(\boldsymbol{c}^t, x)\right)$. We evaluate both strategies empirically and adopt the one that achieves the lower prediction error in subsequent experiments.

## 4 Experiments

We aim to answer the following questions with our experiments: ① *How well can `LucidAtlas` estimate population trends?* (See Sec. 4.2.1.) ② *Can `LucidAtlas` capture heteroscedastic variances across a population?* (See Sec. 4.2.1.) ③ *Is accepting the independence assumption unquestioningly and directly using the interpretations from NAMs appropriate in scientific discovery?* (See Sec. 4.2.2.) ④ *How well can `LucidAtlas` predict responses at time $T_1$ given observations at an earlier time $T_0$?* (Sec. 4.2.3) ⑤ *Can `LucidAtlas` detect out-of-distribution individuals based on uncertainty estimates?* (See Sec. 4.2.3.)

### 4.1 Datasets & Experimental Protocols

Learning a pediatric airway atlas is the primary motivating problem of our work (Hong et al., 2013). The synthetic dataset provides a controlled setting with known ground truth to verify our marginalization approach and heteroscedastic variance estimation. The OASIS Brain Volume dataset (Marcus et al., 2007) serves as a generality check, demonstrating that `LucidAtlas` applies beyond airway analysis to other anatomical domains with different covariates.

We set the regularization weight $\lambda = 1e - 3$ for the loss function Eq. (5) in all experiments. We perform 5-fold cross-validation by patient for all experiments. To assess robustness, we generate five independent synthetic datasets using different random seeds. The Supplementary Material provides more details about these three datasets and our experimental settings in Sec. S.4.

**Synthetic Toy Data.**  We construct a synthetic spatiotemporal dataset with analytically defined signals, deliberately designed to be tractable for conditional analysis. The response variable is a sum of two covariate-modulated spatial functions with heteroscedastic noise. Covariates $c_1$ and $c_2$ are correlated through a known nonlinear transformation, and 30% of the samples have missing values. We generate 200k data points for the training set and 100k for the testing set. The mathematical equations and additional details are available in Sec. S.4.1.

**Pediatric Airway Shape.**  The dataset comprises 358 computed tomography (CT)-derived upper-airway shapes from children with radiographically normal airways, corresponding to 264 patients (34 longitudinal; 230 single visit). We consider three covariates—age, weight, and height; most missingness stems from unrecorded height and truncated scans due to incomplete field-of-view coverage. Each complete airway has 5 anatomical landmarks annotated. Covariates are complete for 263 scans. We aim to construct an atlas of airway cross-sectional area (CSA) and its population distribution, ensuring that CSA increases monotonically with age, weight, and height.

We convert the $k^{\text{th}}$ airway shape into a cross-sectional area (CSA) function $C_k(x)$, which maps normalized depth $x \in [0, 1]$ to CSA values, with $x = 0$ corresponding to the choana and $x = 1$ corresponding to the carina. Based on our discretization, the complete airways have 500 depth-CSA pairs uniformly distributed on the centerline of the airways, while the incomplete ones have $< 500$.

To evaluate the models' ability to detect anatomically abnormal cases, we define Out-of-Distribution (OOD) samples as pediatric airways with subglottic stenosis (SGS) (Zdanski et al., 2016), a clinically relevant airway obstruction. Specifically, we treat the normal cases from the 5-fold cross-validation (CV) test sets as the *in-distribution* data, and use a separate set of 31 SGS scans as the *out-of-distribution* (OOD) group.

**OASIS Brain Volumes.**  Brain segmentations were obtained from the OASIS dataset (Marcus et al., 2007), which includes two subsets: ① A cross-sectional set with 416 subjects aged 18 to 96 years, primarily single-time observations, plus a reliability subset of 20 non-demented subjects rescanned within 90 days. ② A longitudinal set of 150 older adults (60–96 years), totaling 373 imaging sessions.

Our experiments include four covariates: age, socioeconomic status (SES), mini-mental state examination (MMSE), and clinical dementia rating (CDR). The response variable is the normalized whole brain volume (nWBV). *We aim to investigate the relationships between these covariates and brain volume.* Based on prior knowledge, brain volume should not increase with age (Fotenos et al., 2008).

**Comparison Methods.**  We compare our method against several strong baselines for interpretable and uncertainty-aware regression. These include tree-based models such as LightGBM (Ke et al., 2017) and Explainable Boosting Machines (EBM) (Lou et al., 2013), as well as neural additive models (NAM) with ensembling, denoted NAM-Ens (Agarwal et al., 2020). For uncertainty-aware baselines, we consider (i) *PlainMLP*, a multilayer perceptron trained to predict both mean and variance via negative log-likelihood; (ii) GAMLSS, a flexible parametric model for distributional regression (Rigby & Stasinopoulos, 2005); (iii) LA-NAM, a recent extension of NAMs for uncertainty quantification (Bouchiat et al., 2023); and (iv) NAMLSS, a flexible and interpretable distributional regression framework to model each parameter of a

| Method | Pri. | Imp. | Toy Data | OASIS Brain | Pediatric Airway | | | | | |
|---|---|---|---|---|---|---|---|---|---|---|
| | | | | | Overall | choana | epiglottic tip | TVC | subglottis | carina |
| LightGBM | ✗ | ✗ | 6.701±0.033 | 2.931±0.287 | 32.799±1.893 | 33.180±5.697 | 42.380±1.388 | 33.611±1.614 | 17.921±1.767 | 16.940±1.578 |
| EBM | ✗ | ✗ | 6.665±0.008 | 2.889±0.157 | 42.737±0.893 | 40.564±5.351 | 45.653±1.397 | 50.969±2.361 | 35.964±4.451 | 24.414±3.585 |
| NAM-Ens | ✗ | ✗ | 17.884±0.035 | 3.253±0.221 | 34.236±1.305 | 31.723±4.914 | **37.915±2.761** | 44.090±6.600 | 30.090±5.053 | 20.326±2.058 |
| PlainMLP | ✗ | ✗ | **6.065±0.021** | 2.888±0.116 | 33.324±1.724 | 32.980±6.154 | 42.391±1.654 | 33.770±2.100 | 18.858±1.716 | 17.249±3.197 |
| GAMLSS | ✗ | ✗ | 17.936±0.044 | 2.929±0.202 | 33.203±0.524 | 30.515±4.113 | 39.264±3.917 | 44.783±6.719 | 25.891±4.101 | 18.673±1.694 |
| NAMLSS | ✗ | ✗ | 18.514±0.059 | 2.923±0.192 | 35.558±3.114 | 32.063±4.373 | 40.107±3.117 | 42.347±6.827 | 29.099±7.124 | 21.522±6.436 |
| LA-NAM | ✗ | ✗ | 18.530±0.133 | 2.959±0.223 | 37.848±8.311 | 30.171±4.823 | 40.295±3.268 | 53.544±19.188 | 34.633±19.937 | 18.154±2.728 |
| Ours | ✗ | ✗ | **6.060±0.030** | **2.869±0.155** | 34.306±6.705 | 32.776±6.714 | 42.173±2.738 | 41.261±20.076 | 23.138±12.809 | 15.793±2.804 |
| Ours | ✗ | ✓ | 6.066±0.028 | 2.890±0.178 | 30.957±2.522 | 30.517±5.463 | 40.485±2.737 | **31.745±1.135** | 16.729±1.590 | **15.030±2.644** |
| Ours | ✓ | ✗ | 6.065±0.029 | **2.883±0.139** | **30.276±1.628** | **29.589±4.245** | 38.194±3.110 | 32.537±2.876 | **16.682±1.546** | 15.242±2.394 |
| Ours | ✓ | ✓ | 6.085±0.068 | 2.911±0.131 | **29.952±1.594** | **28.973±5.243** | **37.665±3.536** | **31.458±1.460** | **16.270±1.067** | 15.228±2.443 |

Table 2: Quantitative Evaluation of Population Trend Regression on Synthetic and Real Datasets based on Symmetric Mean Absolute Percentage Error (SMAPE, %). We also evaluate with respect to different landmarks. The {TVC, subglottis and carina} landmarks are **important** landmarks for airway obstruction analysis. **Bold red values** indicate the best scores across all methods. **Bold black values** indicate the 2nd best scores of all methods. A ✓ in **Pri.** refers to LucidAtlas incorporating prior knowledge about monotonicity, as illustrated in Sec. 3.2.2. A ✓ in **Imp.** represents using the full dataset for training, including missing values, as illustrated in Sec. 3.5.1. LucidAtlas performs best overall.

| Method | Pri. | Imp. | Toy Data | OASIS Brain | Pediatric Airway | | | | | |
|---|---|---|---|---|---|---|---|---|---|---|
| | | | | | Overall | choana | epiglottic tip | TVC | subglottis | carina |
| PlainMLP | ✗ | ✗ | **-0.463±0.006** | 0.666±0.090 | 1.212±0.572 | 1.749±0.740 | 1.731±0.600 | 0.233±0.207 | -0.090±0.318 | 0.356±0.498 |
| GAMLSS | ✗ | ✗ | 0.776±0.002 | 0.594±0.039 | 0.723±0.094 | 1.497±0.105 | **1.174±0.096** | 0.519±0.090 | 0.117±0.105 | 0.227±0.191 |
| NAMLSS | ✗ | ✗ | 0.887±0.001 | 0.736±0.036 | 0.866±0.146 | 1.587±0.345 | 1.199±0.092 | 0.573±0.277 | 0.398±0.455 | 0.578±0.324 |
| LA-NAM | ✗ | ✗ | 0.964±0.001 | 0.750±0.018 | 0.984±0.046 | **1.389±0.288** | **1.172±0.069** | 0.846±0.016 | 0.802±0.019 | 0.816±0.019 |
| Ours | ✗ | ✗ | **-0.419±0.113** | 0.580±0.046 | 0.646±0.147 | 1.557±0.229 | 1.234±0.046 | 0.156±0.146 | -0.305±0.150 | -0.002±0.172 |
| Ours | ✗ | ✓ | -0.417±0.116 | **0.577±0.054** | **0.603±0.142** | 1.484±0.189 | 1.195±0.090 | **0.048±0.158** | -0.414±0.158 | **-0.051±0.076** |
| Ours | ✓ | ✗ | -0.418±0.113 | **0.576±0.031** | 0.635±0.154 | 1.424±0.190 | 1.313±0.318 | 0.117±0.156 | -0.383±0.158 | -0.035±0.089 |
| Ours | ✓ | ✓ | -0.415±0.115 | 0.579±0.038 | **0.602±0.154** | **1.405±0.256** | 1.212±0.146 | **0.073±0.133** | **-0.415±0.131** | -0.005±0.137 |

Table 3: Quantitative Evaluation of Population Distribution Estimation based on Negative Log-Likelihood (NLL). Our approach achieves the best performance overall.

target distribution (Thielmann et al., 2024). Together, these methods represent the current landscape of interpretable and probabilistic regression approaches.

**Evaluation Metrics.** We evaluated the precision of population trend regression using the Symmetric Mean Absolute Percentage Error (SMAPE) (Makridakis & Hibon, 2000) and the distributional fit with the Negative Log-Likelihood (NLL). For each fold, we report the mean and standard deviation of a trimmed metric that excludes the lowest and highest 5% of errors. For OOD detection, we compute AUC, sensitivity, and specificity by comparing the OOD score (Sec. 3.5.2) between in-distribution and OOD sets.

## 4.2 Discussions

### 4.2.1 Population Trend and Distribution.

***Population Trend.*** Tab. 2, together with the statistical analysis in Tab. S.11, reports the quantitative results for population trend regression. In most cases, LucidAtlas achieves outstanding overall performance.

| Method | Pri. | Imp. | AUC | Sensitivity | Specificity |
|---|---|---|---|---|---|
| PlainMLP | ✗ | ✗ | 0.822±0.029 | 0.819±0.049 | 0.684±0.025 |
| GAMLSS | ✗ | ✗ | 0.758±0.119 | 0.748±0.173 | 0.642±0.100 |
| NAMLSS | ✗ | ✗ | 0.648±0.093 | 0.639±0.108 | 0.577±0.056 |
| Ours | ✗ | ✗ | 0.829±0.066 | 0.852±0.059 | 0.704±0.041 |
| Ours | ✗ | ✓ | 0.842±0.054 | 0.819±0.074 | 0.683±0.040 |
| Ours | ✓ | ✗ | **0.860±0.059** | 0.845±0.092 | 0.699±0.049 |
| Ours | ✓ | ✓ | **0.910±0.014** | **0.903±0.023** | **0.734±0.017** |

Table 4: AUC, Sensitivity, Specificity for Out-of-Distribution Detection on Pediatric Airway Dataset.

| Method | Pediatric Airway | | | |
|---|---|---|---|---|
| | Overall | TVC | subglottis | carina |
| $T_0$ | 17.596 ± 7.892 | 39.081 ± 14.150 | 14.948 ± 9.891 | 18.404 ± 5.634 |
| Pop. | 16.506 ± 4.854 | **37.455 ± 5.580** | 14.183 ± 2.835 | **13.958 ± 3.788** |
| Mean-Shift | **13.346 ± 2.790** | **35.505 ± 7.456** | **10.862 ± 3.280** | 14.835 ± 2.091 |
| Percentile-Preserving | **14.133 ± 2.439** | 38.505 ± 7.217 | **11.218 ± 3.548** | 15.545 ± 3.153 |

Table 5: SMAPE (in %) for Individualized Prediction. $T_0$ in the **Method** column indicates directly using the observation from the initial time point $T_0$ to predict at time $T_1$. **Pop.** indicates using the population trend $f^m(c, x)$ for individualized prediction for $T_1$. The Mean-Shift approach provides the best performance overall and for most landmarks.

| Pediatric Airway | | | | | | | |
|---|---|---|---|---|---|---|---|
| Covariate | Dep. | Overall | choana | epiglottic tip | TVC | subglottis | carina |
| Age | ✓ | **0.589±0.096** | **1.385±0.237** | **1.150±0.075** | **0.107±0.135** | **-0.329±0.142** | **0.045±0.105** |
| Age | ✗ | 0.950±0.100 | 1.487±0.133 | 1.261±0.109 | 0.680±0.111 | 0.525±0.137 | 0.658±0.114 |
| Height | ✓ | **0.602±0.125** | **1.413±0.197** | **1.156±0.109** | **0.086±0.136** | **-0.338±0.144** | **0.022±0.135** |
| Height | ✗ | 0.867±0.132 | 1.475±0.155 | 1.259±0.108 | 0.550±0.140 | 0.358±0.173 | 0.505±0.097 |
| Weight | ✓ | **0.684±0.110** | **1.453±0.212** | **1.169±0.094** | **0.213±0.141** | **-0.134±0.118** | **0.153±0.132** |
| Weight | ✗ | 0.949±0.079 | 1.506±0.143 | 1.221±0.095 | 0.738±0.136 | 0.600±0.216 | 0.764±0.116 |

Table 6: Quantitative Comparison of Different Ways of 1D Covariate Interpretation on Pediatric Airway Dataset. NLL is computed between the marginalized covariate effect and the data distribution. A ✓ in the **Dep.** column indicates that covariate dependence is considered, while ✗ signifies that it is ignored. Accounting for covariate dependence improves alignment between covariate interpretation and the data distribution.

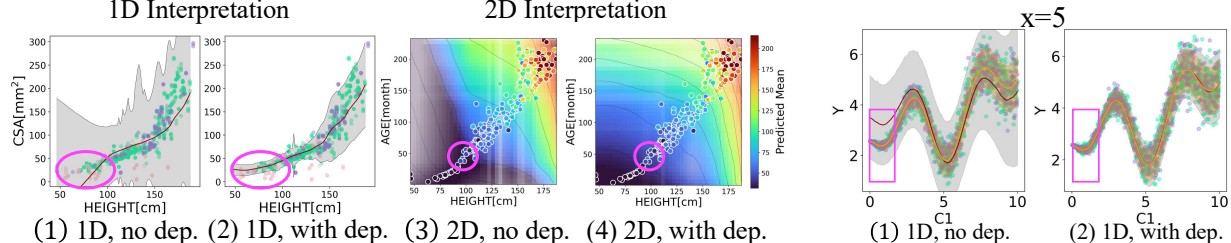

Figure 3: Visualizations of marginalized covariate effects from `LucidAtlas` for CSA distribution at the Subglottis landmark (Pediatric Airway Dataset). (1) 1D effect without covariate dependence; (2) 1D effect with dependence; (3) 2D joint effect without dependence; (4) 2D joint effect with dependence. Green and purple dots show training and testing samples, respectively; red lines indicate the learned trend, with gray shading for ±2× standard deviations. Magenta circles mark regions where accounting for dependence better depicts population trends or uncertainty, underscoring its importance for reliable population-level estimation.

Figure 4: Marginalized covariate effects from `LucidAtlas` on the Synthetic Dataset. The orange band shows ground truth uncertainty, and the gray band shows the estimated uncertainty. (1) Marginalized $p(Y \mid C1)$ without covariate dependence; (2) with dependence modeled.

On the pediatric airway dataset, incorporating monotonicity as a prior and training with data that include missing covariate values, as detailed in Sec. 3.2.2 and Sec. 3.5.1, further improves the performance.

***Population Distribution.*** Tab. 3, together with the statistical analysis in Tab. S.12 quantifies the performance of different methods in estimating population distributions. Compared to the *PlainMLP* baseline, `LucidAtlas` achieves better NLL scores on real-world datasets, highlighting the effectiveness of the additive design in modeling structured uncertainty. Compared to NAMLSS and LA-NAM, `LucidAtlas` further improves performance, demonstrating the importance of explicitly modeling spatial dependence. On the pediatric airway dataset, we observe additional gains from training with partial observations and incorporating prior monotonicity constraints, suggesting that both components are beneficial in real clinical settings.

On the synthetic toy dataset, MLP slightly outperforms our method in terms of NLL. This is expected, as the synthetic signal is intentionally designed to be simple—driven by only two strongly correlated variables—to allow for analytical marginalization as shown in Fig. 4. In this controlled setting, flexible black-box models like MLP are advantaged, though such scenarios are not representative of real-world data complexity.

### 4.2.2 Role of Covariate Dependence.

Tab. 6 (with statistical analysis in Tab. S.13), Fig. 3 and Fig. 4 examine the significance of accounting for covariate dependence in covariate interpretation. Fig. 3 and Fig. 4 illustrate that ignoring the covariate dependence results in a suboptimal estimation of the population distribution, highlighting the need to incorporate the covariate dependence for reliable interpretation. The quantitative results in Tab. 6 further confirm that, covariate interpretation with dependence better aligns with the data distribution. Sec. S.6 includes more results on the synthetic and OASIS Brain Volume dataset.

### 4.2.3 Other Applications.

***Individualized Prediction.*** Tab. 5, together with statistical analysis in Tab. S.13, shows the performance of individualized predictions (to predict for $T_1$ given the observation at $T_0$). The Mean-Shift approach, as described in Sec. 3.5.3, yields the best predictive performance. ***Out-of-Distribution Detection.*** Tab. 4 summarizes the OOD detection performance on the pediatric airway dataset for models that estimate heteroscedastic aleatoric uncertainty. `LucidAtlas` achieves the best results across all metrics. Compared to other competing methods, our method consistently yields superior performance, highlighting the importance of incorporating both spatial structure and additive modeling for reliably identifying pathologically abnormal samples. Notably, the strongest results are obtained when both monotonicity priors and missing-data imputation are employed, emphasizing their contributions to robust OOD detection.

## 5 Limitations and Future Work

While validated on 1D spatial inputs, `LucidAtlas` extends naturally along two axes. Higher-dimensional spatial inputs with scalar outputs (e.g., cortical thickness) require no modification; multi-dimensional outputs (e.g., deformation fields) require extending the predictive distribution to a multivariate Gaussian, while the additive decomposition and marginalization carry over directly. Validating on higher-dimensional benchmarks such as deformation field estimation (Xue et al., 2006; Joshi et al., 2004; Jiao et al., 2023) is planned for future work.

Our experiments use two to four covariates per dataset, which are the most clinically relevant variables in each setting. Scaling to larger covariate spaces is another direction for future work.

`LucidAtlas` models population variances as Gaussian distributions. Expanding beyond Gaussian assumptions, more flexible probabilistic frameworks—such as non-parametric approaches or mixture models—could improve expressiveness and model fits. Identifiability issues arise when covariates are dependent or the latent space is redundant, potentially affecting interpretability (Zhou & Wei, 2020; Siems et al., 2023). Addressing these concerns is crucial for ensuring well-posed solutions.

## 6 Conclusions

We introduced `LucidAtlas`, a probabilistic framework for learning an uncertainty-aware, covariate-disentangled, and individualized atlas representation. Furthermore, we discussed potential risks in using NAMs for interpreting covariates in the presence of covariate dependence and proposed a computationally efficient marginalization approach for arbitrary covariate conditioning. Importantly, our marginalization approach enables a single trained model to produce consistent, arbitrary subset-conditioned predictions and uncertainty estimates. Additionally, `LucidAtlas` enables downstream applications such as individualized temporal prediction and spatially aware out-of-distribution detection. We evaluated our method using a synthetic dataset and two realistic medical datasets, validating its trustworthiness and effectiveness.

### Acknowledgments

This research was, in part, funded by the National Institutes of Health (NIH) under 1R21HL172230-01A1. The views and conclusions contained in this document are those of the authors and should not be interpreted as representing official policies, either expressed or implied, of the NIH.

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

# Supplementary Material for `LucidAtlas`

## S.1 Extended Related Work

**Epistemic Uncertainty versus Aleatoric Uncertainty.** Estimating uncertainty is important to understand the quality of a model fit and to capture variations across a data population. Two different types of uncertainties need to be distinguished: epistemic uncertainty captures model uncertainty whereas aleatoric uncertainty captures uncertainty in the data (Hüllermeier & Waegeman, 2021).

More attention is generally paid to epistemic uncertainties in the context of interpretable models (Wang et al., 2025). NAMs use ensembling to estimate model uncertainties (Agarwal et al., 2020). LA-NAM uses a Laplace approximation for uncertainty estimation (Bouchiat et al., 2023) with NAMs. In atlas construction, aleatoric uncertainty is especially important when individual differences in a dataset are large. Capturing aleatoric uncertainty is crucial in medicine to understand population variations. NAMLSS (Thielmann et al., 2024) can model aleatoric uncertainty by using NAMs to approximate the parameters $\{\theta^{(n)}\}$ of a chosen data distribution (Thielmann et al., 2024), as $\theta^{(n)} = h^{(n)}(\beta^{(n)} + \sum_{i=1}^{N} f_i^{(n)}(c_i))$, where $\theta^{(n)}$ can for example be the mean and variance of Gaussian distributions; $\beta^{(n)}$ denotes the parameter-specific intercept and $f_i^{(n)}$ represents the feature network for the $n$-th parameter for the $i$-th feature. *LucidAtlas incorporates explicit spatial dependence into NAMLSS and leverages the covariate-conditioned distribution for individualized prediction and OOD detection.*

**Heteroscedasticity versus Homoscedasticity.** Distinguishing between homoscedasticity and heteroscedasticity is crucial in statistical analysis, especially for regression models. Homoscedasticity indicates constant variance of random variables, whereas heteroscedasticity indicates that the variance of random variables may differ (Wooldridge et al., 2016). For example, when modeling airway cross-sectional area the population variance may change (increase) with age. `LucidAtlas` supports estimating heteroscedasticity with respect to different locations in an anatomical region and with respect to covariates across a patient population. Many interpretable approaches assume homoscedasticity, e.g., LA-NAM (Bouchiat et al., 2023) assumes homoscedasticity for aleatoric uncertainty in their additive networks. To our knowledge, only NAMLSS considers heteroscedasticity in its additive network design (Thielmann et al., 2024). *LucidAtlas incorporates explicit spatial dependence into NAMLSS and introduces a marginalization approach that enables correct covariate interpretation in the presence of covariate dependence, without relying on the independence assumption underlying standard NAM interpretations.*

**Monotonicity.** Monotonic neural networks ensure that a network's output changes monotonically with respect to certain inputs. Research has focused on two lines of approaches: architectures such as Deep Lattice Networks (You et al., 2017) that guarantee monotonicity but may lack expressiveness, and heuristic methods such as Certified Monotonic Neural Networks (Liu et al., 2020) that use regularization but can be computationally expensive. Recent advancements, including Constrained Monotonic Neural Networks (Runje & Shankaranarayana, 2023), aim to balance monotonicity, expressiveness, and efficiency. Additionally, research in normalizing flows (De Cao et al., 2020) has contributed to developing monotonic functions in neural networks to ensure invertibility. Expressive monotonic neural networks (Kitouni et al., 2023) are constructed using Lipschitz-constrained neural networks, ensuring monotonicity by design while preserving expressiveness. *We use the Lipschitz-constrained neural networks to ensure monotonicity in LucidAtlas to follow prior / domain knowledge.*

**Disentangled Representation Learning.** Disentangled representation learning (DRL) has been explored in a variety of domains, including computer vision (Shoshan et al., 2021; Ding et al., 2020; Zhang et al., 2018b;a; Xu et al., 2021; Yang et al., 2020), natural language processing (John et al., 2018), and medical image analysis (Chartsias et al., 2019; Bercea et al., 2022).

Medical data is typically associated with various covariates which should be taken into account during analyses. Taking (Chu et al., 2022) as an example, when observing a tumor's progression, it is difficult to know whether the variation of a tumor's progression is due to time-varying covariates or due to treatment

| Notations | Explanations |
|---|---|
| $y$ | Observed variable, i.e., target variable to model |
| $\boldsymbol{c}$ | A vector containing all $N$ covariates, e.g, $\boldsymbol{c} = [age, weight, ...]$ |
| $\boldsymbol{c}_{\mathcal{S}}$ | A vector containing the covariates in set $\mathcal{S}$ |
| $f^m(\boldsymbol{c}, x)$ or $f^m$ | Prediction of mean population trend given $\boldsymbol{c}$ at location $x$ |
| $f_i^m(c_i, x)$ or $f_i^m$ | Additive effects predicted from $i^{th}$ subnetwork $f_i$ for mean |
| $f^v(\boldsymbol{c}, x)$ or $f^v$ | Prediction of population variance given $\boldsymbol{c}$ at location $x$ |
| $f_i^v(c_i, x)$ or $f_i^v$ | Additive effects predicted from $i^{th}$ subnetwork $f_i$ for variance |
| $g^m(\boldsymbol{c}_{\mathcal{S}})$ | The predicted mean of $\boldsymbol{c}$ given $\boldsymbol{c}_{\mathcal{S}}$ |
| $g^v(\boldsymbol{c}_{\mathcal{S}})$ | The predicted covariance matrix of $\boldsymbol{c}$ given $\boldsymbol{c}_{\mathcal{S}}$ |
| $p(y \mid \boldsymbol{c}_{\mathcal{S}}, x)$ | Marginalized covariate effects: how $\boldsymbol{c}_{\mathcal{S}}$ affect $y$ at location $x$ |
| $\mathrm{E}[y \mid \boldsymbol{c}_{\mathcal{S}}, x]$ | The expectation of $y$ when $\boldsymbol{c}_{\mathcal{S}}$ and $x$ are fixed |
| $\mathrm{Var}(y \mid \boldsymbol{c}_{\mathcal{S}}, x)$ | The variance of $y$ when $\boldsymbol{c}_{\mathcal{S}}$ and $x$ are fixed |

Table S.7: Illustrations of the Notations.

effects. Therefore, being able to disentangle different effects is highly useful for a representation to promote understanding and to be able to quantify the effect of covariates on observations. *LucidAtlas disentangles covariate effects in terms of their contribution to population trends and uncertainties.*

**Explainable Artificial Intelligence.** The goal of eXplainable Artificial Intelligence (XAI) is to provide human-understandable explanations for the decisions and actions of AI models. Various approaches to XAI have been proposed, including counterfactual inference (Berrevoets et al., 2021; Moraffah et al., 2020; Thiagarajan et al., 2020; Chen et al., 2022), attention maps (Zhou et al., 2016; Jung & Oh, 2021; Woo et al., 2018), feature importance (Arik & Pfister, 2021; Ribeiro et al., 2016; Agarwal et al., 2020), and instance retrieval (Crabbe et al., 2021). A neural additive model (NAM) (Agarwal et al., 2020; Jiao et al., 2023) is an important XAI method that achieves interpretability through a linear combination of neural networks, each focusing on a *single* input feature. NAISR pioneers the use of NAMs for modeling medical shapes to enable scientific discoveries in the medical domain (Jiao et al., 2023); however, it does not account for heteroskedasticity in its shape representation and does not consider uncertainties. *LucidAtlas extends this concept by integrating NAMs to construct an atlas that captures population trends and uncertainties with spatial dependence.*

## S.2 Method

### S.2.1 Notations

Tab. S.7 shows the notations used in this paper.

### S.2.2 Expanded Discussion on the Toy Example

Assuming $c_1$ and $c_2$ are covariates that influence the observed result $y$, a NAM fits well whose subnetworks capture $f_1(c_1) = \sin(c_1)$ and $f_2(c_2) = c_2$ and thus approximate $y$ with $y = f(c_1, c_2) + \epsilon = f_1(c_1) + f_2(c_2) + \epsilon$, where $\epsilon$ is Gaussian noise with mean zero.

If we want to interpret the population trend of $y$ with only $c_1$, we need to marginalize $c_2$ out as

$$
\begin{aligned}
F_1(c_1) &= \int [f_1(c_1) + f(c_2)] p(c_2 \mid c_1) \, \mathrm{d}c_2 \\
&= \underbrace{f_1(c_1)}_{\text{Interpretation from NAMs}} + \underbrace{\int f_2(c_2) p(c_2 \mid c_1) \, \mathrm{d}c_2}_{\text{Interpretation from Dependence: } := h_1(c_1)} = \sin(c_1) + \int c_2 p(c_2 \mid c_1) \, \mathrm{d}c_2
\end{aligned}
\tag{S.19}
$$

where $h_1(c_1)$ measures how the dependence between $c_1$ and $c_2$ influences the marginalization $F_1(c_1)$. We can see from Eq. (S.19) that $F_1(c_1)$ is composed of the interpretation from the NAM's subnetwork plus the interpretation from the dependence between $c_1$ and $c_2$ as $h_1(c_1)$.

If we want to interpret the population trend of $y$ with only $c_2$, we need to marginalize $c_1$ out as

$$
\begin{aligned}
F_2(c_2) &= \int [f_2(c_2) + f(c_1)]p(c_1 \mid c_2)\,\mathrm{d}c_1 \\
&= \underbrace{f_2(c_2)}_{\text{Interpretation from NAMs}} + \underbrace{\int f_1(c_1)p(c_1 \mid c_2)\,\mathrm{d}c_1}_{\text{Interpretation from Dependence: } := h_2(c_2)} = c_2 + \int \sin(c_1)p(c_1 \mid c_2)\,\mathrm{d}c_1 \,.
\end{aligned} \tag{S.20}
$$

**If $c_1$ and $c_2$ are *independent*,**

$$
\begin{aligned}
h_1(c_1) &= \int f_2(c_2)p(c_2 \mid c_1)\,\mathrm{d}c_2 = \int f_2(c_2)p(c_2)\,\mathrm{d}c_2 = \mathrm{E}_{p(c_2)}[f_2(c_2)] = constant \,, \\
h_2(c_2) &= \int f_1(c_1)p(c_1 \mid c_2)\,\mathrm{d}c_1 = \int f_1(c_1)p(c_1)\,\mathrm{d}c_1 = \mathrm{E}_{p(c_1)}[f_1(c_1)] = constant \,.
\end{aligned} \tag{S.21}
$$

Thus,

$$
F_1(c_1) = \sin(c_1) + \mathrm{E}[c_2] \,, \quad F_2(c_2) = c_2 + \mathrm{E}[\sin(c_1)] \tag{S.22}
$$

which means the marginalization is the actual covariate disentanglement $f_i^m(c_i)$ plus a constant.

**If $c_1$ and $c_2$ are *dependent*,** $h(c_1)$ is a function of $c_1$ which is controlled by the dependence between $c_1$ and $c_2$.

For example, assume that the relationship between $c_1$ and $c_2$ is at one extreme of dependence that $c_2$ is a deterministic function of $c_1$ as

$$
c_2 = \exp(c_1) \,. \tag{S.23}
$$

Then

$$
F_1(c_1) = \sin(c_1) + \int c_2 p(c_2 \mid c_1)\,\mathrm{d}c_2 = \sin(c_1) + \exp(c_1) \tag{S.24}
$$

$$
F_2(c_2) = c_2 + \int \sin(c_1)p(c_1 \mid c_2)\,\mathrm{d}c_1 = c_2 + \sin(\log(c_2)) \,. \tag{S.25}
$$

Therefore, modeling the dependence between $c_1$ and $c_2$ is crucial when using either covariate to interpret the population trend.

In summary, the structurally separated covariate effects of NAMs, combined with those effects contributed by covariate dependence, shape human-understandable explanations that align with population distributions. *While ignoring potential dependence in NAMs may not impact prediction performance, it can result in ambiguous or misleading interpretations when analyzing population trends.*

### S.2.3 Marginalization Approach

### S.2.3.1 Marginalized Covariate Effects

The section introduces our marginalization approach, which enables covariate analysis conditioning on any arbitrary subset of covariates. The observation $y$ can be formulated as

$$
y = f^m(\boldsymbol{c}, x) + f^v(\boldsymbol{c}, x) \cdot \epsilon \,, \ \epsilon \sim \mathcal{N}(0, 1) \,. \tag{S.26}
$$

We expand the two-covariate case in Sec. 3.3 to the multi-covariate setting. Let $\mathcal{S} \subseteq \{1, \ldots, N\}$ denote a set of covariate indices. For any such index set, we define $\boldsymbol{c}_{\mathcal{S}} = (c_i)_{i \in \mathcal{S}}$ as the subvector of $\boldsymbol{c} = (c_1, \ldots, c_N)$

containing the covariates indexed by $\mathcal{S}$. Thus, $\mathcal{S}$ specifies which covariates are selected, while $\boldsymbol{c}_{\mathcal{S}}$ denotes their corresponding values as an ordered vector. Note that while $\mathcal{S}$ is a set, the notation $\boldsymbol{c}_{\mathcal{S}}$ explicitly preserves the indexing and order of the selected covariates. For simplicity, we denote the complement by $\boldsymbol{c}_{-\mathcal{S}} = (c_i)_{i \notin \mathcal{S}}$. We now aim to derive the mean and variance of the conditional distribution $p(y \mid \boldsymbol{c}_{\mathcal{S}}, x)$, as $f^m(\boldsymbol{c}_{\mathcal{S}}, x)$ and $f^v(\boldsymbol{c}_{\mathcal{S}}, x)$ respectively.

**Mean of $p(y \mid \boldsymbol{c}_{\mathcal{S}}, x)$.** We derive the conditional mean of $p(y \mid \boldsymbol{c}_{\mathcal{S}}, x)$ by marginalizing over the complementary covariates $\boldsymbol{c}_{-\mathcal{S}}$. Formally, it is defined as

$$f^m(\boldsymbol{c}_{\mathcal{S}}, x) = \mathbb{E}_{\boldsymbol{c}_{-\mathcal{S}} \mid \boldsymbol{c}_{\mathcal{S}}} \big[ y \mid \boldsymbol{c}_{\mathcal{S}}, x \big]. \tag{S.27}$$

Since $y$ is modeled as $y = f^m(\boldsymbol{c}, x) + f^v(\boldsymbol{c}, x) \cdot \epsilon$ with $\epsilon \sim \mathcal{N}(0, 1)$, we obtain

$$= \mathbb{E}_{\boldsymbol{c}_{-\mathcal{S}} \mid \boldsymbol{c}_{\mathcal{S}}} \big[ f^m(\boldsymbol{c}, x) + f^v(\boldsymbol{c}, x) \cdot \epsilon \big]. \tag{S.28}$$

Since $\mathbb{E}[\epsilon] = 0$, the noise term vanishes:

$$= \mathbb{E}_{\boldsymbol{c}_{-\mathcal{S}} \mid \boldsymbol{c}_{\mathcal{S}}} \big[ f^m(\boldsymbol{c}, x) \big]. \tag{S.29}$$

This expectation can be written as a multidimensional integral over those covariates not in $\mathcal{S}$:

$$= \int f^m(\boldsymbol{c}, x) \, p(\boldsymbol{c}_{-\mathcal{S}} \mid \boldsymbol{c}_{\mathcal{S}}) \, \mathrm{d}\boldsymbol{c}_{-\mathcal{S}}. \tag{S.30}$$

Using the additive structure $f^m(\boldsymbol{c}, x) = \sum_{i=1}^{N} f_i^m(c_i, x) + b^m(x)$, we can separate the contributions from covariates in $\mathcal{S}$ and those outside:

$$= \sum_{i \in \mathcal{S}} f_i^m(c_i, x) + \underbrace{\int \left( \sum_{i \notin \mathcal{S}} f_i^m(c_i, x) \right) p(\boldsymbol{c}_{-\mathcal{S}} \mid \boldsymbol{c}_{\mathcal{S}}) \, \mathrm{d}\boldsymbol{c}_{-\mathcal{S}}}_{:= H} + b^m(x). \tag{S.31}$$

Here, each term $f_i^m(c_i, x)$ represents the contribution from the $i$-th additive subnetwork of `LucidAtlas`, corresponding to covariate $c_i$. The term $H$ in Eq. (S.31) accounts for the contributions of the dependence between the covariates.

Thanks to the additive structure of $f^m(\boldsymbol{c}, x)$, this expectation decomposes into a sum of univariate integrals. In particular, $H$ simplifies as

$$\begin{aligned} H &= \sum_{i \notin \mathcal{S}} \int f_i^m(c_i, x) \, p(\boldsymbol{c}_{-\mathcal{S}} \mid \boldsymbol{c}_{\mathcal{S}}) \, dc_{-\mathcal{S}} \overset{(a)}{=} \sum_{i \notin \mathcal{S}} \int f_i^m(c_i, x) \left( \int p(\boldsymbol{c}_{-\mathcal{S} \setminus \{i\}} \mid c_i, \boldsymbol{c}_{\mathcal{S}}) \, d\boldsymbol{c}_{-\mathcal{S} \setminus \{i\}} \right) p(c_i \mid \boldsymbol{c}_{\mathcal{S}}) \, dc_i \\ &\overset{(b)}{=} \sum_{i \notin \mathcal{S}} \int f_i^m(c_i, x) \, p(c_i \mid \boldsymbol{c}_{\mathcal{S}}) \, dc_i \,. \end{aligned} \tag{S.32}$$

In (a) we apply the chain rule $p(\boldsymbol{c}_{-\mathcal{S}} \mid \boldsymbol{c}_{\mathcal{S}}) = p(\boldsymbol{c}_{-\mathcal{S} \setminus \{i\}} \mid c_i, \boldsymbol{c}_{\mathcal{S}}) \, p(c_i \mid \boldsymbol{c}_{\mathcal{S}})$. We marginalize all covariates in $\boldsymbol{c}_{-\mathcal{S}}$ except for $c_i$ (denoted as $\boldsymbol{c}_{-\mathcal{S} \setminus \{i\}}$). In (b) we use $\int p(\boldsymbol{c}_{-\mathcal{S} \setminus \{i\}} \mid c_i, \boldsymbol{c}_{\mathcal{S}}) \, d\boldsymbol{c}_{-\mathcal{S} \setminus \{i\}} = 1$.

This step exploits the fact that each $f_i^m(c_i, x)$ only depends on a single covariate, allowing us to marginalize out the remaining dimensions inside the joint conditional distribution. As a result, computing the conditional mean $f^m(\boldsymbol{c}_{\mathcal{S}}, x)$ only requires access to the univariate conditionals $p(c_i \mid \boldsymbol{c}_{\mathcal{S}})$, which simplifies computations (discussed in Sec. 3.4.2).

This leads to the final expression for the conditional expectation:

$$f^m(\boldsymbol{c}_{\mathcal{S}}, x) = \sum_{i \in \mathcal{S}} f_i^m(c_i, x) + \sum_{i \notin \mathcal{S}} \int f_i^m(c_i, x) p(c_i \mid \boldsymbol{c}_{\mathcal{S}}) \mathrm{d}c_i + b^m(x). \tag{S.33}$$

**Variance of $p(y \mid \boldsymbol{c}_\mathcal{S}, x)$.** Now, we derive the conditional variance of $p(y \mid \boldsymbol{c}_\mathcal{S}, x)$. The *law of total variance* is $\mathrm{Var}(Y) = \mathbb{E}[\mathrm{Var}(Y \mid X)] + \mathrm{Var}(\mathbb{E}[Y \mid X])$, which states that the total variance of a random variable $Y$ can be broken into two parts: ① the **expected variance of $Y$ given $X$**, which represents how much $Y$ fluctuates around its mean for each specific value of $X$; and ② **the variance of the expected value of $Y$ given $X$**, which measures how much the conditional mean itself varies as $X$ changes.

In our case, we apply this to the predictive distribution $p(y \mid \boldsymbol{c}_\mathcal{S}, x)$, where only a subset $\boldsymbol{c}_\mathcal{S}$ of covariates is known, and the rest of the covariates $\boldsymbol{c}_{-\mathcal{S}}$ are marginalized. Conditioning on $\boldsymbol{c}_\mathcal{S}$, we write:

$$f^v(\boldsymbol{c}_\mathcal{S}, x) = \mathrm{Var}(y \mid \boldsymbol{c}_\mathcal{S}, x) = \underbrace{\mathbb{E}_{\boldsymbol{c}_{-\mathcal{S}}|\boldsymbol{c}_\mathcal{S}}\left[\mathrm{Var}(y \mid \boldsymbol{c}_{-\mathcal{S}}, \boldsymbol{c}_\mathcal{S}, x)\right]}_{①:=\tilde{\sigma}_E^2(\boldsymbol{c}_\mathcal{S}, x)} + \underbrace{\mathrm{Var}_{\boldsymbol{c}_{-\mathcal{S}}|\boldsymbol{c}_\mathcal{S}}\left(\mathbb{E}[y \mid \boldsymbol{c}_{-\mathcal{S}}, \boldsymbol{c}_\mathcal{S}, x]\right)}_{②:=\tilde{\sigma}_V^2(\boldsymbol{c}_\mathcal{S}, x)} .$$

$$(S.34)$$

Using the additive structure $f^v(\boldsymbol{c}, x) = \sum_{i=1}^N f_i^v(c_i, x) + b^v(x)$, we compute:

$$\tilde{\sigma}_E^2(\boldsymbol{c}_\mathcal{S}, x) = \mathbb{E}_{\boldsymbol{c}_{-\mathcal{S}}|\boldsymbol{c}_\mathcal{S}}\left[\mathrm{Var}(y \mid \boldsymbol{c}_{-\mathcal{S}}, \boldsymbol{c}_\mathcal{S}, x)\right] = \mathbb{E}_{\boldsymbol{c}_{-\mathcal{S}}|\boldsymbol{c}_\mathcal{S}}\left[f^v(\boldsymbol{c}, x)\right] = \int f^v(\boldsymbol{c}, x) p(\boldsymbol{c}_{-\mathcal{S}} \mid \boldsymbol{c}_\mathcal{S}) d\boldsymbol{c}_{-\mathcal{S}}$$

$$= \sum_{i \in \mathcal{S}} f_i^v(c_i, x) + \sum_{i \notin \mathcal{S}} \int f_i^v(c_i, x) \, p(c_i \mid \boldsymbol{c}_\mathcal{S}) \, dc_i + b^v(x).$$

$$(S.35)$$

We now explain the second term, $\tilde{\sigma}_V^2(\boldsymbol{c}_\mathcal{S}, x)$, which corresponds to the variance of the conditional mean function $\mathbb{E}[y \mid \boldsymbol{c}_{-\mathcal{S}}, \boldsymbol{c}_\mathcal{S}, x] = f^m(\boldsymbol{c}, x)$. Because we are conditioning on $\boldsymbol{c}_\mathcal{S}$, the functions $f_i^m(c_i, x)$ for $i \in \mathcal{S}$ are deterministic, so only the terms for $i \notin \mathcal{S}$ contribute to variance. Therefore:

$$\tilde{\sigma}_V^2(\boldsymbol{c}_\mathcal{S}, x) = \mathrm{Var}_{\boldsymbol{c}_{-\mathcal{S}}|\boldsymbol{c}_\mathcal{S}}\left(\sum_{i \notin \mathcal{S}} f_i^m(c_i, x)\right).$$

$$(S.36)$$

To compute this, we use the identity for the variance of a sum (Loève, 1977):

$$\mathrm{Var}\left(\sum_i Z_i\right) = \sum_i \mathrm{Var}(Z_i) + \sum_{i \neq j} \mathrm{Cov}(Z_i, Z_j).$$

$$(S.37)$$

Applying this to our context gives:

$$\tilde{\sigma}_V^2(\boldsymbol{c}_\mathcal{S}, x) = \mathrm{Var}_{\boldsymbol{c}_{-\mathcal{S}}|\boldsymbol{c}_\mathcal{S}}\left(\mathbb{E}[y \mid \boldsymbol{c}, x]\right) = \mathrm{Var}_{\boldsymbol{c}_{-\mathcal{S}}|\boldsymbol{c}_\mathcal{S}}\left(f^m(\boldsymbol{c}, x)\right) = \mathrm{Var}_{\boldsymbol{c}_{-\mathcal{S}}|\boldsymbol{c}_\mathcal{S}}\left(\sum_{i \in \mathcal{S}} f_i^m(c_i, x) + \sum_{i \notin \mathcal{S}} f_i^m(c_i, x) + b^m(x)\right)$$

$$= \mathrm{Var}_{\boldsymbol{c}_{-\mathcal{S}}|\boldsymbol{c}_\mathcal{S}}\left(\sum_{i \notin \mathcal{S}} f_i^m(c_i, x)\right) = \sum_{i \notin \mathcal{S}} \underbrace{\mathrm{Var}_{c_i|\boldsymbol{c}_\mathcal{S}}\left(f_i^m(c_i, x)\right)}_{③} + \sum_{\substack{j_1 \neq j_2 \\ j_1, j_2 \notin \mathcal{S}}} \underbrace{\mathrm{Cov}_{(c_{j_1}, c_{j_2})|\boldsymbol{c}_\mathcal{S}}\left(f_{j_1}^m(c_{j_1}, x), f_{j_2}^m(c_{j_2}, x)\right)}_{④}$$

$$(S.38)$$

③ is the variance of $f_i^m(c_i, x)$ under the conditional distribution $p(c_i \mid \boldsymbol{c}_\mathcal{S})$:

$$③ := \mathrm{Var}_{c_i|\boldsymbol{c}_\mathcal{S}}\left(f_i^m(c_i, x)\right) = \int \left(f_i^m(c_i, x) - \tilde{\mu}_i(\boldsymbol{c}_\mathcal{S}, x)\right)^2 p(c_i \mid \boldsymbol{c}_\mathcal{S}) \, dc_i, \quad \tilde{\mu}_i(\boldsymbol{c}_\mathcal{S}, x) = \int f_i^m(c_i, x) p(c_i \mid \boldsymbol{c}_\mathcal{S}) dc_i. \quad (S.39)$$

④ is the conditional covariance between $f_{j_1}^m(c_{j_1}, x)$ and $f_{j_2}^m(c_{j_2}, x)$ under the joint conditional distribution $p(c_{j_1}, c_{j_2} \mid \boldsymbol{c}_\mathcal{S})$:

$$④ := \mathrm{Cov}_{(c_{j_1}, c_{j_2})|\boldsymbol{c}_\mathcal{S}}\left(f_{j_1}^m(c_{j_1}, x), f_{j_2}^m(c_{j_2}, x)\right) = \iint f_{j_1}^m(c_{j_1}, x) f_{j_2}^m(c_{j_2}, x) \, p(c_{j_1}, c_{j_2} \mid \boldsymbol{c}_\mathcal{S}) \, dc_{j_1} \, dc_{j_2} - \tilde{\mu}_{j_1}(\boldsymbol{c}_\mathcal{S}, x) \, \tilde{\mu}_{j_2}(\boldsymbol{c}_\mathcal{S}, x).$$

$$(S.40)$$

From Eqs. (S.35) to (S.40), all integrals are performed over $c_i$ or $(c_{j_1}, c_{j_2})$, conditioned on $\boldsymbol{c}_\mathcal{S}$, and are evaluated under the corresponding multivariate distributions.

Eqs. (S.35) to (S.40) reveal that, rather than sampling from the full covariate space, it suffices to sample from conditional distributions such as $p(c_i \mid \boldsymbol{c}_\mathcal{S})$ and $p(c_{j_1}, c_{j_2} \mid \boldsymbol{c}_\mathcal{S})$ in order to compute the marginalized predictive distribution $p(y \mid \boldsymbol{c}_\mathcal{S}, x)$. In other words, our proposed approach allows marginalization over arbitrary subsets of covariates without requiring model retraining.

## S.3 Network Architecture

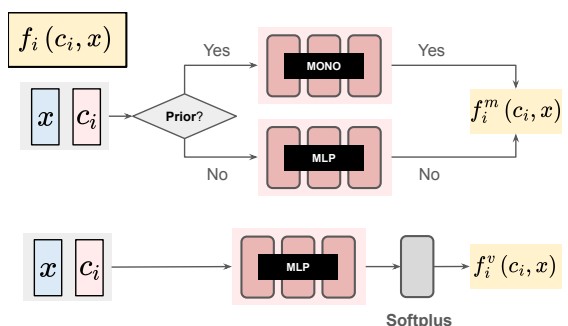
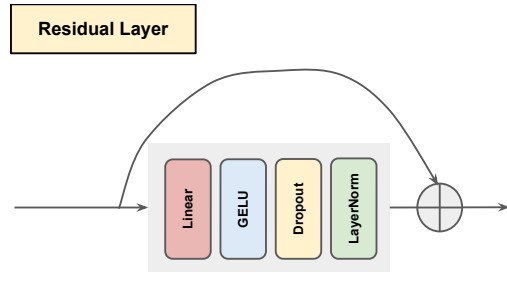

Figure S.5: Network architecture of additive subnetwork $f_i$ in `LucidAtlas`.

Figure S.6: Network architecture of residual layers.

Fig. S.5 shows the network architecture of the additive subnetwork $f_i$, which receives the anatomical location $x$ and covariate $c_i$ to predict the additive contribution $f_i^m(c_i, x)$ and $f_i^v(c_i, x)$ to the mean and variance for the distributional parameters $f^m(\boldsymbol{c}, x)$ and $f^v(\boldsymbol{c}, x)$ respectively. Specifically, if there is prior knowledge, we use a monotonic neural network (Kitouni et al., 2023) as the backbone to predict $f_i^m(c_i, x)$; if there is no prior knowledge, we use an MLP to predict $f_i^m(c_i, x)$. Considering that the variance should be a number $\geq 0$, a *softplus* activation layer is used at the output of the MLP for $f_i^v(c_i, x)$ to ensure $f_i^v(c_i, x)$ is a non-negative number.

We use residual layers in all MLPs. Figure S.6 illustrates the residual layer architecture used in $f_i$ as well as in the subnetworks $\phi(\cdot)$, $\rho(\cdot)$, and $\varphi(\cdot)$ within $g(\cdot)$.

## S.4 Datasets

### S.4.1 Toy Dataset

We construct a synthetic spatiotemporal dataset with analytically defined signals, deliberately designed to be tractable for conditional analysis. The generation process is defined as follows:

Let $x \in [0, 10]$ denote a one-dimensional spatial coordinate, and let $c_1, c_2$ denote two continuous covariates. The outcome variable $y$ is given by:

$$y = f_1(x, c_1) + f_2(x, c_2) + \varepsilon(x, c_1, c_2)$$

where:

$$f_1(x, c_1) = \sin(0.15(c_1 + 1)x) + \cos(0.3(c_1 + 1)x), \quad f_2(x, c_2) = \exp(0.1(c_2 + x + 1))$$

$$\varepsilon(x, c_1, c_2) \sim \mathcal{N}\left(0, (\sigma_1(x, c_1) + \sigma_2(x, c_2))^2\right)$$

with:

$$\sigma_1(x, c_1) = 0.05 \cdot x \cdot (0.1 \cdot c_1), \quad \sigma_2(x, c_2) = 0.05 \cdot x \cdot (\exp(0.1 \cdot c_2) - 1)$$

To introduce covariate dependence, $c_2$ is generated conditionally from $c_1$ via:

$$c_2 = \log(c_1 + 1) \cdot 4 + \xi, \quad \xi \sim \mathcal{N}(0, \sigma_\xi^2)$$

where we use $\sigma_\xi = 0$ for a deterministic mapping.

Additionally, we simulate missingness by setting 30% of $c_2$ values to NaN uniformly at random, enabling evaluation under partial observation.

| # observations | 1 | 2 | 3 | 4 | 5 | 6 | 7 | 9 | 11 |
|---|---|---|---|---|---|---|---|---|---|
| # patients | 230 | 12 | 6 | 8 | 3 | 2 | 1 | 1 | 1 |

Table S.8: Number of patients for a given number of observations for the pediatric airway dataset. For example, the 1st column indicates that there are 230 patients who were only observed once.

### S.4.2 OASIS Brain

The Open Access Series of Imaging Studies (OASIS) is a project aimed at making MRI data sets of the brain freely available to the scientific community (Marcus et al., 2007).

The OASIS Brain dataset we use is publicly available in a preprocessed form [2]. The OASIS Brain dataset consists of two sets, i.e.,

1. **A Cross-Sectional MRI Dataset (416 Subjects, Ages 18–96).** 100 of the included subjects are over the age of 60 and have been clinically diagnosed with very mild to moderate Alzheimer's disease (AD). Additionally, a reliability data set is included containing 20 nondemented subjects imaged on a subsequent visit within 90 days of their initial session.

2. **A Longitudinal MRI Dataset in Nondemented and Demented Older Adults (150 Subjects, Ages 60–96).** This set consists of a longitudinal collection of 150 subjects aged 60 to 96. Each subject was scanned on two or more visits, separated by at least one year for a total of 373 imaging sessions. For each subject, 3 or 4 individual T1-weighted MRI scans obtained in single scan sessions are included. The subjects are all right-handed and include both men and women. 72 of the subjects were characterized as nondemented throughout the study. 64 of the included subjects were characterized as demented at the time of their initial visits and remained so for subsequent scans, including 51 individuals with mild to moderate Alzheimer's disease. Another 14 subjects were characterized as nondemented at the time of their initial visit and were subsequently characterized as demented at a later visit.

Our experiments include four covariates: age, socioeconomic status (SES), mini-mental state examination (MMSE), and clinical dementia rating (CDR). The outcome variable is normalized whole brain volume (nWBV), which is a scalar.

*We aim to investigate the relationships between these covariates and brain volume.* Based on prior knowledge, the atlas brain volume should not increase with age (Fotenos et al., 2008; Hedman et al., 2012; Guimarães et al., 2012).

### S.4.3 Pediatric Airway

The airway shapes are extracted from computed tomography (CT) images. The real CT images are from children ranging in age from 1 month to ~19 years old. Acquiring CT images is costly. Further, CT uses ionizing radiation which should be avoided, especially in children, due to cancer risks. Hence, it is difficult to acquire such CTs for many children. Instead, the data was acquired by serendipity from children who received CTs for reasons other than airway obstructions (e.g., because they had cancer) (Jiao et al., 2023). This also explains why it is difficult to acquire longitudinal data. E.g., one of the patients has 11 timepoints because a very sick child was scanned 11 times.

The pediatric airway dataset includes 230 cross-sectional observations (where a patient was only imaged once) and 34 longitudinal observations. 176 patients (i.e., 263 shapes) have all three covariates (age, weight, height) and 11 annotated anatomical landmarks. 5 landmarks are located on the upper airway section for this experiment. Errors in the shapes $\{\mathcal{S}^k\}$ may arise from image segmentation error, differences in head positioning, missing parts of the airway shapes due to incomplete image coverage, and dynamic airway

---

[2]https://www.kaggle.com/datasets/jboysen/mri-and-alzheimers

| P- | 0 | 10 | 20 | 30 | 40 | 50 | 60 | 70 | 80 | 90 | 100 |
|---|---|---|---|---|---|---|---|---|---|---|---|
| age | 1.00 | 23.00 | 55.00 | 71.00 | 89.00 | 111.00 | 129.00 | 161.00 | 179.00 | 199.00 | 233.00 |
| m-vol | 4.56 | 16.84 | 29.53 | 28.91 | 27.31 | 70.90 | 71.23 | 43.34 | 78.63 | 102.35 | 113.84 |

Table S.9: Visualization and demographic information of our 3D airway shape dataset. Shapes of $\{0, 10, 20, 30, 40, 50, 60, 70, 80, 90, 100\}$-th age percentiles are plotted with their covariates (age/month) printed in the table. M-vol (measured volume) is the volume ($cm^3$) of the gold standard shapes based on the actual imaging.

deformations due to breathing. Tab. S.8 shows the distribution of the number of observations across patients. Most of the patients in the dataset only have one observation; only 22 patients have $\geq 3$ observation times. Tab. S.9 shows the shapes and demographic information at different age percentiles for the whole data set. Similar to the OASIS Brain dataset, the time span of the longitudinal data for each patient is far shorter than the time span across the entire dataset.

### S.4.3.1 Data Preparation for Pediatric Airway Atlas

The image processing pipeline includes three steps: 1) automatic airway segmentation from CT images; 2) airway representation with a centerline and cross sections.

**Airway Segmentation.** A deep learning-based approach is used for automatic upper airway segmentation from CT images. The segmentation model is trained in two steps. The first step predicts the segmentation using a coarse version of the scans. The second step makes the segmentation prediction on original images. This step takes in the image as input, but also uses the first step prediction as an additional input. Each step is implemented as a U-Net (Özgün Çiçek et al., 2016; Ronneberger et al., 2015).

The automatic segmentation model is developed based on a dataset containing 68 pairs of airway CT images and their corresponding manual segmentations.

**Centerline and Cross Sections.** The pediatric airway dataset is constructed by extracting 358 airway geometries from CT images with our automatic segmentation approach. The upper airways, like any tube-like structures, can be approximated by a centerline with cross sections (Hong et al., 2013). Following the approach in (Hong et al., 2013), the airway centerline is inferred based on the heat distribution along the airway provided by solving Laplace's equation. The iso-surfaces of heat values are extracted from the Laplace solution and the centerlines are considered as the centers of the iso-surfaces. Cross sections are cut from the airway geometry using planes that are orthogonal to the tangent of the centerline.

**Pediatric Airway Atlas Construction.** Similar as the approach in (Hong et al., 2013), the cross-sectional area is considered as the airway's main feature. For each point on the centerline, it has a distance $x$ from the choana which is normalized to 1 over the length of the airway, and a cross-sectional area $y$. The 1D function for airway geometry is the curve $c(x)$ that smoothly passes through all these points on the centerline, as $y = c(x)$.

The airway curves are aligned based on five key anatomic landmarks $\{\boldsymbol{p}_i\}$: choana, epiglottic tip, true vocal cord (TVC), subglottis, and carina.

Each landmark $\boldsymbol{p}_i = (p_{ix}, p_{iy}, p_{iz})$ is projected onto the centerline to obtain the corresponding depth $x_i$ along the centerline. For example, the depth of choana $x_{choana}$ should be at 0 while the depth of carina $x_{\text{carina}}$

| Model | LucidAtlas | | | NAMLSS | PlainMLP |
|---|---|---|---|---|---|
| | $f_i^m$ or $f_i^v$ | | $g_i$ | $f_i^m$ or $f_i^v$ | *PlainMLP* |
| | Monotonic | MLP | $\phi(\cdot),\ \varphi(\cdot),\ \rho(\cdot)$ | MLP | |
| Hidden Layers | $[512, 512, 512]$ | $[512, 512, 512]$ | $[512, 512, 512]$ | $[512, 512, 512]$ | $[512, 512, 512]$ |
| Activation | GroupSort | GeLU | GeLU | GeLU | GeLU |
| Learning Rate | 1e-3 | 1e-3 | 1e-4 | 1e-3 | 1e-3 |
| # Epochs | 500 | | | | |
| Dropout | 0.2 | | | | |
| Others | Adam optimizer, CosineAnnealingLR, Earlystopping | | | | |

Table S.10: Hyperparameter Settings for Comparison Methods. GroupSort is introduced in (Kitouni et al., 2023).

should be at 1. For each landmark, there is a mean position $\overline{\boldsymbol{p}}_i = (\bar{p}_{ix}, \bar{p}_{iy}, \bar{p}_{iz})$ and the mean depth $\bar{d}_i$ of that landmark over all cases.

A landmark-based curve registration approach (Hong et al., 2013) is used to estimate a piecewise linear warping function $h_k(\cdot)$ for each curve $c_k(\cdot)$, which is strictly monotonic and places the landmark points for a particular subject $k$ at the mean location of these landmarks in the atlas, $x_i = h_k(\bar{x}_i)$. With the constructed warping functions, curves can then be resampled to the normalized coordinate system with $C_k(x) = c_k(h_k(x))$.

### S.4.3.2 Evaluation Metrics

The Symmetric Mean Absolute Percentage Error (SMAPE) (Makridakis & Hibon, 2000) is used to evaluate regression accuracy in capturing population-level trends, defined as

$$\text{SMAPE} = \frac{100\%}{n} \sum_{i=1}^{n} \frac{|y_i - \hat{y}_i|}{(|y_i| + |\hat{y}_i|)/2}, \tag{S.41}$$

where $y_i$ is the ground truth and $\hat{y}_i$ is the predicted value. Unlike traditional MAPE, SMAPE is symmetric and bounded, making it robust when the true values approach zero.

The Negative Log-Likelihood (NLL) measures how well the modeled distribution aligns with the true data distribution. For each fold, we report the mean and standard deviation of a trimmed metric, i.e., the lowest and highest 5% of errors are excluded, to mitigate the influence of outliers and better reflect performance stability. The corresponding untrimmed results are provided in Sec. S.6. For OOD detection, we compute AUC, sensitivity, and specificity by comparing the OOD score proposed in Sec.3.5.2 between the in-distribution sets and the OOD group.

## S.5 Experimental Setup

Tab. S.10 illustrates the hyperparameter settings of our approach, NAMLSS (Thielmann et al., 2024) and the *PlainMLP* comparison. For our approach and all comparison methods, we use 15% of the training data set by patient as a validation set for early stopping. The batch size for the Pediatric Airway Dataset is set to 1024, while for the OASIS Brain dataset it is set to 32. For other comparison methods, we use their publicly available implementation, which we describe in the following.

**NAM.** We use the official PyTorch implementation of NAM [3]. We evaluate the NAM using feature networks with three hidden layers (to keep consistent with hyperparameter settings in Table. S.10), each containing 512 hidden units. A dropout rate of 0.2 is applied, and the ensemble consists of 20 learners. We use 15% of the training data set by patient as a validation set for early stopping. All other experimental settings follow the recommended or default configurations.

---

[3]https://github.com/lemeln/nam/tree/main?tab=readme-ov-file

**Explainable Boosting Machine.** The explainable boosting machine (EBM) is an open-source Python implementation [4] of the gradient-boosting GAM that is available as a part of the InterpretML library (Lou et al., 2013; Nori et al., 2019). We use 15% of the training data set by patient as a validation set for early stopping. We use the default hyperparameter setting because we did not find a significant improvement when tuning the hyperparameters.

**LightGBM.** LightGBM is a gradient boosting framework that uses tree-based learning algorithms (Ke et al., 2017). We use the open-source implementation [5]. We use 15% of the training data set by patient as a validation set for early stopping. We find that the recommended or default configurations work well.

**Hardware.** The deep learning models are trained on a single NVIDIA GeForce RTX 3090 GPU and an Intel(R) Xeon(R) Gold 6226R CPU.

## S.6   More Results and Visualizations

Tab. S.11 and Tab. S.12 present the statistical analyses for modeling the population average trend and distribution, using sample-level Wilcoxon rank-based tests on SMAPE and NLL, respectively. On the pediatric airway dataset, our method achieves the best overall performance and is statistically superior on most landmarks. On the synthetic toy and OASIS brain volume datasets, it significantly outperforms most competing approaches.

Tab. S.13 reports the statistical analyses for different ways of 1D covariate interpretations on the Pediatric Airway dataset. We compute sample-level Wilcoxon signed-rank tests to compare NLL scores between the **with-dependence** and **without-dependence** models across all covariates and anatomical landmarks. The results consistently show highly significant p-values, indicating that explicitly modeling covariate dependence substantially improves the alignment between marginalized covariate effects and the underlying data distribution. These findings confirm the necessity of accounting for covariate dependence when interpreting pediatric airway development.

Tab. S.14 summarizes the statistical analysis for individualized prediction on the Pediatric Airway dataset. One-sided Wilcoxon signed-rank tests are conducted on the SMAPE scores to compare different prediction strategies. The Mean-Shift method achieves the best overall performance as well as the best accuracy at most landmarks with statistical significance.

Tab. S.15 compares the joint marginalized covariate effects of multiple covariates on the pediatric airway dataset, with and without accounting for covariate dependence. The results confirm that incorporating covariate dependence leads to interpretations that better align with the underlying data distribution.

Fig. S.7 visualizes the marginalized covariate effects for single and multiple covariates, with and without covariate dependence. In this case, incorporating or not incorporating covariate dependence does not make a big difference because age and MMSE are not correlated. Tab. S.17 and Tab. S.18 evaluate joint marginalized covariate effects for single and multiple covariates, respectively, on the OASIS brain volume dataset, evaluated with and without accounting for covariate dependence. The results indicate that incorporating covariate dependence yields interpretations that more accurately reflect the underlying data distribution. The individualized prediction performance in Tab. S.19 remains the same between the mean-shift and percentile-preserving approaches, because the uncertainty is homoscedastic as shown in Fig. S.7.

Tab. S.16 evaluates the impact of accounting for covariate dependence in marginalization for the synthetic dataset. The results highlight that considering the dependence is crucial for accurately interpreting the effects of individual covariates in neural additive models.

---

[4] https://github.com/interpretml/interpret/tree/3e810552f7fcae641bf6bd945f10c66bf56c424b
[5] https://lightgbm.readthedocs.io/en/latest/index.html

| Method | Pri. | Imp. | Toy Data | OASIS Brain | Pediatric Airway | | | | | |
|--------|------|------|----------|-------------|---------|--------|----------------|-----|-----------|--------|
| | | | | | Overall | choana | epiglottic tip | TVC | subglottis | carina |
| LightGBM | ✗ | ✗ | <0.001 | 0.438 | <0.001 | <0.001 | <0.001 | <0.001 | <0.001 | <0.001 |
| EBM | ✗ | ✗ | <0.001 | 0.732 | <0.001 | <0.001 | <0.001 | <0.001 | <0.001 | <0.001 |
| NAM-Ens | ✗ | ✗ | <0.001 | <0.001 | <0.001 | <0.001 | 0.018 | <0.001 | <0.001 | <0.001 |
| PlainMLP | ✗ | ✗ | 0.993 | 0.808 | <0.001 | <0.001 | <0.001 | <0.001 | <0.001 | <0.001 |
| GAMLSS | ✗ | ✗ | <0.001 | 0.680 | <0.001 | <0.001 | <0.001 | <0.001 | <0.001 | <0.001 |
| NAMLSS | ✗ | ✗ | <0.001 | 0.314 | <0.001 | <0.001 | <0.001 | <0.001 | <0.001 | <0.001 |
| LA-NAM | ✗ | ✗ | <0.001 | 0.356 | <0.001 | <0.001 | <0.001 | <0.001 | <0.001 | <0.001 |
| Ours | ✗ | ✗ | 0.977 | 0.912 | <0.001 | <0.001 | <0.001 | <0.001 | <0.001 | 0.066 |
| Ours | ✗ | ✓ | 0.984 | 0.815 | <0.001 | <0.001 | <0.001 | 0.686 | 0.022 | 0.970 |
| Ours | ✓ | ✗ | 0.898 | 0.807 | <0.001 | <0.001 | <0.001 | 0.091 | 0.264 | 0.941 |
| Ours | ✓ | ✓ | N/A (self) | N/A (self) | N/A (self) | N/A (self) | N/A (self) | N/A (self) | N/A (self) | N/A (self) |

Table S.11: Statistical Analysis of SMAPE Scores for Each Comparison Method against `LucidAtlas` (**Ours_full**). Reported values are one-sided Wilcoxon rank-based $p$-values computed on sample-level SMAPE scores, testing whether `LucidAtlas` yields significantly lower error than each method ($H_1$: `LucidAtlas` (**Ours_full**) < comparison method). Smaller $p$-values indicate stronger evidence of superiority, with $p < 0.05$ denoting statistically significant improvement.

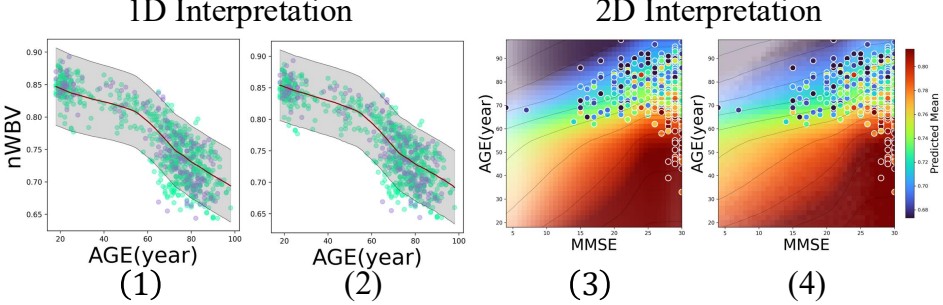

Figure S.7: Visualizations of Marginalized Covariate Effects from `LucidAtlas` for OASIS Brain Volume Dataset. (1) 1D selective marginal effect without accounting for covariate dependence; (2) 1D selective marginal effect with covariate dependence modeled; (3) 2D joint selective marginal effect without modeling covariate dependence; (4) 2D joint selective marginal effect incorporating covariate dependence.

| Method | Pri. | Imp. | Toy Data | OASIS Brain | Pediatric Airway | | | | | |
|---|---|---|---|---|---|---|---|---|---|---|
| | | | | | Overall | choana | epiglottic tip | TVC | subglottis | carina |
| PlainMLP | ✗ | ✗ | >0.999 | 0.200 | <0.001 | <0.001 | <0.001 | <0.001 | <0.001 | <0.001 |
| GAMLSS | ✗ | ✗ | <0.001 | 0.025 | <0.001 | <0.001 | <0.001 | <0.001 | <0.001 | <0.001 |
| NAMLSS | ✗ | ✗ | <0.001 | <0.001 | <0.001 | <0.001 | 0.376 | <0.001 | <0.001 | <0.001 |
| LA-NAM | ✗ | ✗ | <0.001 | <0.001 | <0.001 | 0.055 | 0.020 | <0.001 | <0.001 | <0.001 |
| Ours | ✗ | ✗ | 0.886 | 0.009 | <0.001 | <0.001 | <0.001 | <0.001 | <0.001 | 0.188 |
| Ours | ✗ | ✓ | 0.770 | 0.855 | <0.001 | <0.001 | >0.999 | >0.999 | <0.001 | 0.717 |
| Ours | ✓ | ✗ | 0.550 | 0.027 | <0.001 | <0.001 | 0.991 | <0.001 | 0.009 | >0.999 |
| Ours | ✓ | ✓ | N/A (self) | N/A (self) | N/A (self) | N/A (self) | N/A (self) | N/A (self) | N/A (self) | N/A (self) |

Table S.12: Statistical Analysis of NLL Scores for Each Comparison Method against `LucidAtlas` (**Ours_full**). Reported values are one-sided Wilcoxon rank-based $p$-values computed on sample-level NLL scores, testing whether `LucidAtlas` yields significantly lower error than each method ($H_1$: `LucidAtlas` (**Ours_full**) < comparison method). Smaller $p$-values indicate stronger evidence of superiority, with $p < 0.05$ denoting statistically significant improvement.

| Pediatric Airway | | | | | | | |
|---|---|---|---|---|---|---|---|
| Covariate | Dep. | Overall | choana | epiglottic tip | TVC | subglottis | carina |
| Age | ✓ | N/A (self) | N/A (self) | N/A (self) | N/A (self) | N/A (self) | N/A (self) |
| Age | ✗ | <0.001 | <0.001 | <0.001 | <0.001 | <0.001 | <0.001 |
| Height | ✓ | N/A (self) | N/A (self) | N/A (self) | N/A (self) | N/A (self) | N/A (self) |
| Height | ✗ | <0.001 | <0.001 | <0.001 | <0.001 | <0.001 | <0.001 |
| Weight | ✓ | N/A (self) | N/A (self) | N/A (self) | N/A (self) | N/A (self) | N/A (self) |
| Weight | ✗ | <0.001 | <0.001 | <0.001 | <0.001 | <0.001 | <0.001 |

Table S.13: Statistical Analysis of 1D Covariate Interpretations on the Pediatric Airway Dataset. Entries report one-sided Wilcoxon signed-rank $p$-values comparing sample-level NLL between the **with-dependence** and **without-dependence** models for each covariate and landmark. The $H_1$- hypothesis is whether the **with-dependence** model achieves significantly lower NLL. A ✓ in the **Dep.** column indicates that covariate dependence is modeled, whereas ✗ denotes that it is ignored. Smaller $p$-values indicate stronger evidence that modeling covariate dependence improves alignment between marginalized covariate effects and the data distribution; in particular, $p < 0.05$ denotes statistically significant improvement.

| Method | Pediatric Airway | | | |
|---|---|---|---|---|
| | Overall | TVC | subglottis | carina |
| $T_0$ | <0.001 | <0.001 | <0.001 | <0.001 |
| Pop. | <0.001 | <0.001 | <0.001 | 0.916 |
| Mean-Shift | N/A (self) | N/A (self) | N/A (self) | N/A (self) |
| Percentile-Preserving | <0.001 | <0.001 | <0.001 | <0.001 |

Table S.14: Statistical Analysis of Individualized Prediction on the Pediatric Airway Dataset. Entries report one-sided Wilcoxon signed-rank $p$-values comparing sample-level SMAPE between our **Mean-Shift** approach and other prediction methods. We test whether the Mean-Shift method yields significantly lower SMAPE than other approaches. In the **Method** column, $T_0$ denotes directly using the initial observation at $T_0$ to predict at $T_1$, and **Pop.** denotes using the population trend $f^m(c, x)$ for individualized prediction at $T_1$. The Mean-Shift approach yields the best overall performance.

| Pediatric Airway | | | | | | | |
|---|---|---|---|---|---|---|---|
| Covariate | Dep. | Overall | choana | epiglottic tip | TVC | subglottis | carina |
| [Age, Height] | ✓ | **0.579±0.128** | **1.402±0.235** | **1.167±0.108** | **0.060±0.124** | **-0.409±0.125** | **-0.011±0.119** |
| [Age, Height] | ✗ | 0.900±0.231 | 1.471±0.176 | 1.306±0.161 | 0.529±0.263 | 0.242±0.317 | 0.535±0.260 |
| [Age, Weight] | ✓ | **0.607±0.112** | **1.399±0.250** | **1.180±0.110** | **0.103±0.121** | **-0.362±0.124** | **0.020±0.131** |
| [Age, Weight] | ✗ | 0.840±0.079 | 1.427±0.198 | 1.215±0.115 | 0.593±0.101 | 0.398±0.161 | 0.467±0.102 |
| [Weight, Height] | ✓ | **0.631±0.149** | **1.451±0.225** | **1.197±0.137** | **0.104±0.149** | **-0.330±0.148** | **0.022±0.148** |
| [Weight, Height] | ✗ | 0.755±0.118 | 1.447±0.174 | 1.207±0.137 | 0.375±0.106 | 0.144±0.212 | 0.401±0.193 |

Table S.15: Quantitative Comparison of Different Approaches for (N–1)D Covariate Interpretation on the Pediatric Airway Dataset. For example, [Age, Height] in the **Covariate** column indicates that only age and height are used to interpret the data distribution. NLL is computed between the marginalized covariate interpretation and the data distribution. Accounting for covariate dependence improves alignment between covariate interpretation and the data distribution.

| | | Toy Data | | | | | | |
|---|---|---|---|---|---|---|---|---|
| Feat. | Dep. | Overall | 1 | 3 | 5 | 7 | 9 | 10 |
| C1 | ✓ | **-0.420±0.113** | **-1.763±0.079** | **-0.684±0.066** | **-0.153±0.072** | **0.149±0.084** | **0.407±0.077** | **0.550±0.061** |
| C1 | ✗ | 0.580±0.134 | 0.344±0.177 | 0.334±0.187 | 0.490±0.148 | 0.756±0.084 | 1.000±0.094 | 1.110±0.068 |
| C2 | ✓ | **-0.419±0.112** | **-1.764±0.080** | **-0.684±0.066** | **-0.152±0.072** | **0.150±0.083** | **0.407±0.076** | **0.553±0.061** |
| C2 | ✗ | 0.907±0.268 | 0.598±0.388 | 0.928±0.259 | 0.951±0.240 | 1.062±0.218 | 1.140±0.179 | 1.249±0.180 |

Table S.16: Quantitative Comparison of Different Ways of 1D marginalized covariate on Synthetic Dataset. NLL is computed between the marginalized covariate effect and the data distribution. Accounting for covariate dependence improves alignment between marginalized covariate effects and the data distribution.

| OASIS Brain Volume | | |
|---|---|---|
| Covariate | Dep. | Overall |
| AGE | ✓ | 0.738±0.090 |
| AGE | ✗ | **0.724±0.054** |
| CDR | ✓ | **0.966±0.073** |
| CDR | ✗ | 1.310±0.081 |
| MMSE | ✓ | **1.008±0.092** |
| MMSE | ✗ | 1.280±0.077 |
| SES | ✓ | **1.065±0.025** |
| SES | ✗ | 1.238±0.045 |

Table S.17: Quantitative Comparison of Different Ways of 1D Covariate Interpretation on for the OASIS Brain Volume Dataset. NLL is computed between the marginalized covariate interpretation and the data distribution. Accounting for covariate dependence improves alignment between covariate interpretation and the data distribution.

| OASIS Brain Volume | | |
|---|---|---|
| Covariate | Dep. | Overall |
| [Age, MMSE, CDR] | ✓ | **0.604±0.054** |
| [Age, MMSE, CDR] | ✗ | 0.617±0.068 |
| [Age, SES, CDR] | ✓ | **0.624±0.063** |
| [Age, SES, CDR] | ✗ | 0.644±0.069 |
| [Age, SES, MMSE] | ✓ | **0.630±0.056** |
| [Age, SES, MMSE] | ✗ | 0.664±0.046 |
| [Age, MMSE, CDR] | ✓ | **0.948±0.063** |
| [Age, MMSE, CDR] | ✗ | 1.280±0.078 |

Table S.18: Quantitative Comparison of Different Ways of (N-1)D Covariate Interpretation on for the OASIS Brain Volume Dataset. NLL is computed between the marginalized covariate interpretation and the data distribution. Accounting for covariate dependence improves alignment between covariate interpretation and the data distribution.

| Method | OASIS Brain |
|---|---|
| Pop. | 1.517±0.319 |
| T0 | 3.077±0.465 |
| Mean-Shift | **1.188±0.254** |
| Percentile-Preserving | **1.188±0.254** |

Table S.19: Symmetric Mean Absolute Percentage Error (in %) for Individualized Prediction on OASIS Brain Volume Dataset. $T_0$ in the **Method** column indicates directly using the observation from the initial time point $T_0$ to predict at time $T_1$. **Pop.** indicates using the population trend $f^m(c, x)$ for individualized prediction for $T_1$. The Mean-Shift approach provides the best performance for both datasets and for most landmarks.

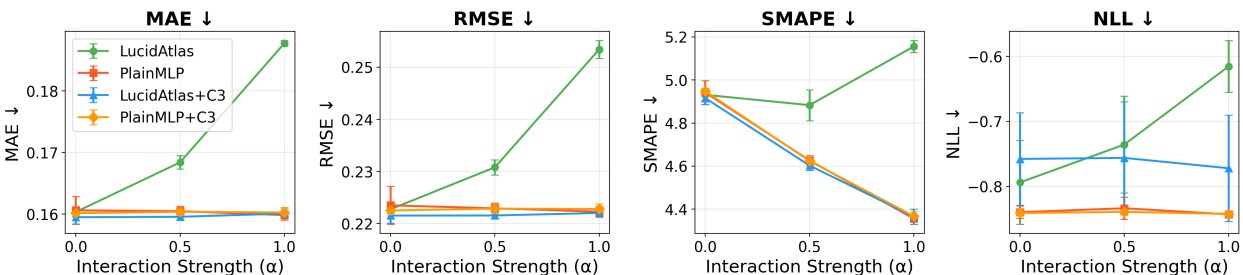

Figure S.8: Robustness to non-additive interactions with independent covariates $c_1, c_2 \sim U(0, 10)$. We vary the interaction strength $\alpha \in \{0, 0.5, 1.0\}$ and compare `LucidAtlas` and PlainMLP, with and without an explicit interaction feature $c_3 = c_1 \cdot c_2$. ↓ indicates lower is better. `LucidAtlas` degrades as $\alpha$ increases, but adding $c_3$ recovers performance comparable to `PlainMLP`.

## S.7 Robustness to Non-Additive Interactions

**Setup.** To evaluate robustness when the additive assumption is violated, we construct a non-additive synthetic dataset. The data-generating process follows Sec. S.4.1 with two independent covariates $c_1, c_2 \sim U(0, 10)$ and an additional interaction term:

$$y = f_1(x, c_1) + f_2(x, c_2) + \alpha \cdot \frac{c_1 c_2 \sin(0.3x)}{50} + \varepsilon, \tag{S.42}$$

where $\alpha \in \{0, 0.5, 1.0\}$ controls the interaction strength. We compare four models: `LucidAtlas`, `PlainMLP`, and their variants augmented with an explicit interaction feature $c_3 = c_1 \cdot c_2$ as a third covariate (`LucidAtlas+C3`, `PlainMLP+C3`). The goal is to examine (i) how `LucidAtlas` degrades under non-additive signals, and (ii) whether providing the interaction feature explicitly can recover performance.

**Regression Performance.** We report Mean Absolute Error (MAE), Root Mean Squared Error (RMSE), and Symmetric Mean Absolute Percentage Error (SMAPE) in Fig. S.8. In terms of MAE and RMSE, `LucidAtlas` without $c_3$ degrades as $\alpha$ increases, while `PlainMLP` remains stable, confirming that the additive model cannot capture $c_1 \cdot c_2$ interactions. `LucidAtlas+C3` recovers performance comparable to `PlainMLP` across all three metrics. SMAPE decreases for all methods as $\alpha$ grows; this is because the interaction term enlarges the dynamic range of $y$, so the denominator $(|y| + |\hat{y}|)/2$ grows faster than the absolute error, reducing the relative percentage metric.

**Distribution Modeling.** Negative Log-Likelihood (NLL) results show that `LucidAtlas` without $c_3$ worsens with increasing $\alpha$. Adding the explicit interaction feature $c_3$ stabilizes NLL across all $\alpha$ values, with only a minor gap compared to `PlainMLP`. This indicates that the heteroscedastic variance estimation also benefits from the explicit interaction feature.

**Summary.** While `LucidAtlas` inherits the additive limitation of NAMs, this can be effectively mitigated by incorporating known interaction features as additional covariates—preserving per-covariate interpretability while accommodating non-additive effects.

