# OpenReview forum: "$\texttt{LucidAtlas}$: Learning Uncertainty-Aware,  Covariate-Disentangled, Individualized Atlas Representations"
_TMLR — Accepted by TMLR_

### Review · Reviewer_qZ92 · 2025-11-19

**Summary Of Contributions:**

The paper extends NAMLSS by modeling both spatial structure and covariate interactions. It introduces a marginalization framework that captures how combinations of covariates jointly influence the learned anatomical atlas, enabling more accurate and interpretable explanations. The method also handles missing covariates by learning conditional relationships between observed and unobserved variables. It leverages its uncertainty estimates to identify regions where a subject’s anatomy deviates from population norms. The claims in the paper are supported by experiments over real and synthetic datasets.

**Audience:**

Yes

**Audience Explanation:**

The paper considers an interesting problem of designing interpretable by design models for regression by considering the spatial and covariate interactions in the problem.

**Claims And Evidence:**

Yes

**Claims Explanation:**

The paper provides a convincing modelling of spatial structure using appropriate the model architecture and loss function. Interpretability was enhanced by considering the covariate dependence explicitly. Experimental results further support the claims made in the paper.

**Requested Changes:**

While I overall liked the paper but I believe the writing should be improved to make the paper accessible to a broader audience  working in explainability/interpretability and who may not understand the specifics of the biological problem considered in this paper. Some of my suggestions are as follows.

1. It was not completely clear when reading the introduction what spatial dependence is being talked about. First, as the word populations were mentioned several times, I thought it is about people data from the different locations. But that was not the case. It might good to put up a clear example like that of the airway to say where this spatial dependence come from.

2. I understood from the paper that existing NAMs do not account for spatial dependence. But what about other techniques? I am not asking for new experiments here but a bit of discussion on how this problem has been tackled so far.

3. There is a bit of disconnect moving from Section 3.4.1 to Section 3.4.2. DeepSets come in without any motivation or introduction. You are modelling g(c_s) which was not defined before that.

---

> ### Author Response · Authors · 2026-02-20
> **Response to Reviewer qZ92 for LucidAtlas**
>
> We thank the reviewer for the thoughtful feedback on improving accessibility for a broader audience. We have revised the writing throughout the paper to better connect the technical formulation to concrete examples, as detailed below.
>
> ---
>
> **Q1: Unclear what "spatial dependence" means in the Introduction.**
>
> We have rewritten **Section 1: Introduction** to make this concept concrete before it is formalized. The revised structure first introduces the type of atlas (scalar measurements at spatially corresponding locations), then grounds it in a specific setting (tubular anatomies where geometry is summarized as cross-sectional area along a centerline). With this context established, the Spatial-level challenge paragraph can now convey "spatial dependence" concretely: different anatomical locations along the airway (e.g., subglottis vs. carina) exhibit distinct population trends and covariate effects. We believe this progression — atlas type → concrete application → spatial challenge — resolves the ambiguity the reviewer identified.
>
> ---
>
> **Q2: How do other (non-NAM) methods handle spatial dependence?**
>
> We have added a **Spatial Dependence Modeling** paragraph in **Section 2: Related Work**. The key distinction is how covariate effects relate to spatial location. In Gaussian process regression (Rasmussen, 2003), spatial correlations are captured through a kernel $k(x, x')$, but covariate effects enter as global fixed effects that are the same at every location. Similarly, conditional autoregressive (CAR) models (Banerjee et al., 2003\) model spatial dependence through neighbor structure, but again covariate effects are global coefficients shared across all spatial units. In both cases, covariate effects do not vary by location, and there is no per-covariate decomposition. $\\texttt{LucidAtlas}$ addresses this by letting each covariate subnetwork $f\_i(c\_i, x)$ take the spatial coordinate as input, so that covariate effects are simultaneously nonlinear, per-covariate disentangled, and spatially varying.
>
> ---
>
> **Q3: Abrupt transition from §3.4.1 to §3.4.2; DeepSets introduced without motivation.**
>
> We have fixed this problem by adding a transition sentence at the beginning of **Section 3.4.2**:  *The marginalized covariate effects derived in Sec. 3.4.1 require evaluating conditional distributions such as $p(c\_i \\mid c\_S)$ and $p(c\_{j\_1}, c\_{j\_2} \\mid c\_S)$ for arbitrary subsets $S$. This calls for a flexible model $g(\\boldsymbol{c}\_{\\mathcal{S}})$ of that can model conditional distributions such as $p(c\_i \\mid \\boldsymbol{c}\_{\\mathcal{S}})$ and $p(c\_{j\_1}, c\_{j\_2} \\mid \\boldsymbol{c}\_{\\mathcal{S}})$ with arbitrary subsets $\\mathcal{S} \\subseteq \\{1,\\dots,N\\}$ of observed covariates as input.*
>
> ---
>
> \[1\] Rasmussen, Carl Edward. "Gaussian processes in machine learning." *Summer school on machine learning*. Berlin, Heidelberg: Springer Berlin Heidelberg, 2003\. 63-71.
> \[2\] Banerjee, Sudipto, Bradley P. Carlin, and Alan E. Gelfand. *Hierarchical modeling and analysis for spatial data*. Chapman and Hall/CRC, 2003\.

---

### Review · Reviewer_PF9C · 2025-12-09

**Summary Of Contributions:**

The paper presents a unified methodology to construct biological atlases that involve spatiotemporal modeling. The main approach is based on combining Neural Additive Models with spatially aware neural subnetworks to model population trends, heteroscedasticity, and spatial dependency simultaneously. The methodology involves a dependence-aware marginalization approach using DeepSet neural nets to disentangle covariate effects. This approach addresses the limitations of methods that assume covariate independence.

The main contributions of the paper are:
* Neural Additive Models (NAM) rely on the assumption of covariate independence, which can result in ambiguous interpretations. LucidAtlas introduces dependence-aware marginalization as a solution to mitigate these problems.
* Previous models do not consider spatial variation, while LucidAtlas incorporates this variable in the modeling.
* The proposed approach enables individualized predictions and out of distribution detection.
* The empirical results show improved performance in the experimental evaluation that includes synthetic data and two real datasets.

**Audience:**

Yes

**Audience Explanation:**

Modeling population variation in an interpretable way that facilitates individualized predictions in a spatial atlas is an important research problem in statistics and machine learning. The applications in biomedical sciences are also of wide interest. Both machine learning researchers and biomedical researchers will be interested in the methods and applications.

**Broader Impact Concerns:**

No broader impact concerns.

**Claims And Evidence:**

No

**Claims Explanation:**

The paper introduces various desirable properties of an atlas that can support reasoning of population and individual effects in spatial biological data. These properties are experimentally validated, primarily in one dataset. The experimental evidence includes:
* Evaluation of population distribution estimation, using negative log likelihood.
* Sensitivity and specificity of out of distribution detection.
* Analysis of personalized predictions.
* Analysis of the effect of considering dependencies with covariates vs ignoring dependencies.
* Interpretable visualizations of the trends and covariate effects.

The evidence seems accurate, as far as this reviewer can tell. The main problem is that the provided evidence is difficult to follow and interpret in context. The narrative of the paper presents statistical arguments, modeling decisions, and interpretations of the proposed approach in the abstract, leaving a large disconnect between the main focus of the study (Pediatric Airway Dataset) and the theoretical aspects. This disconnect leads to confusion for the reader, in the following aspects:

* **Type of spatial atlas being modeled**. Atlases are extensively studied in biomedical research, and the focus of this paper is not the construction of the spatial atlas. The spatial processing is independently conducted with existing methods, resulting in an observed outcome per patient, which is obscured in the presentation of the method. Using the "word" atlas twice in the title creates high expectations with respect to the manipulation of the spatial data, which is reduced to a single readout in practice throughout the paper.
* **Covariates type and number**. The presented integrated model introduces prior information (e.g., monotonicity) and other inductive biases that are problem dependent. However, it is unclear why those are introduced until later in the paper when the results and datasets are described. The formulation is generic in terms of the number of covariates, but in practice the evaluation only considers two. This is also a mismatch in expectations between the presented method and the actual problem.

**Requested Changes:**

Please, consider the following changes to the paper to facilitate the interpretation of the methods and the results:

* **Describe the application first**. The Pediatric Airway Dataset is the main interest of the authors, as stated in the manuscript. It is an important application and it should be clear from the beginning. Make an effort to introduce the application as the central motivation to develop the methods, and to facilitate the association between variables, covariates, and other inductive bias decisions to the grounding problem.
* **Use the problem to describe the method**. While the formulation is generic, the presentation can be improved by reminding the reader why the choices are made in the context of the Pediatric Airway Dataset. Covariates like age and height, and the fact that these are monotonic make sense only at the end, making the interpretation difficult in the middle of the paper.
* **Use the other datasets as additional analysis**. The main results are focused on the Pediatric Airway problem, and most of the validations and discussions are around it. The other datasets are only used marginally, which suggest that they only serve as an additional test case, rather than the central topic of study. I do not suggest removing them, but rather giving them the right place. Currently, many of the applications discussed in the paper are not tested in these datasets. They are used to test generality, but they also confuse the reader because other analyses are not presented.

---

> ### Author Response · Authors · 2026-02-20
> **Response to Reviewer PF9C for LucidAtlas**
>
> We thank the reviewer for the detailed critique. We note that the reviewer's "No" on whether claims are supported refers to the narrative presentation rather than technical correctness — as the reviewer states, "the evidence seems accurate... the main problem is that the provided evidence is difficult to follow and interpret in context." We have addressed this through the restructuring described below.
>
> ---
>
> **Key weakness: Disconnect between generic formulation and the primary application; "atlas" expectations; covariate number mismatch.**
>
> We appreciate this comment. *We have revised the **Section 1: Introduction** to lead with the clinical application before the technical framework, and added concrete clinical motivation where design choices are introduced (see Q1).* Our revised Introduction clarifies that "atlas" here refers to a population-level statistical model of scalar measurements, not the spatial correspondence step itself (see Q2). Regarding covariate number, we acknowledge the gap between the generic formulation and the small number of covariates in our experiments (2–4), which reflects clinical relevance rather than a framework limitation (see Q4).
>
> ---
>
> **Q1: Describe the application first; use the problem to motivate method choices.**
>
> We have restructured **Section 1: Introduction** (please see the updated manuscript) to lead with the clinical problem: Paragraph 1 introduces scalar atlas construction as a general paradigm, Paragraph 2 narrows to tubular anatomies as a natural setting with pediatric airway as the primary example, followed by four application-grounded challenges and contributions.
>
> Beyond the Introduction, we have added concrete clinical motivation throughout the Method section where design choices are introduced — e.g., the monotonicity prior (§3.2.2) now opens with the example that airway CSA increases with age and height; the DeepSets transition (§3.4.2) now explains why marginalization requires flexible conditioning on arbitrary covariate subsets, a need arising from clinical scenarios where only partial covariate information is available.
>
> ---
>
> **Q2: "Atlas" in the title sets expectations for 2D/3D spatial processing.**
>
> We have clarified in the **Section 1: Introduction** that once spatial correspondence is established, local geometry can be summarized as a scalar quantity, e.g., cross sectional area for airways or cortex thickness for brains, and our framework models how these scalars vary across space and covariates. We have also added a discussion in **Section 5: Limitations and Future Work** on extensions to higher-dimensional settings, distinguishing higher-dimensional inputs (no modification needed) from higher-dimensional outputs (requires multivariate Gaussian), and noting concrete benchmarks for future validation.
>
> ---
>
> **Q3: Other datasets are used marginally; their roles are unclear.**
>
> We have added explicit role descriptions in **Section 4.1: Datasets & Experimental Protocols** the synthetic dataset validates correctness of marginalization and heteroscedastic variance estimation against known ground truth; ***the pediatric airway dataset is the primary clinical application*** with full pipeline evaluation; OASIS serves as a generality check demonstrating applicability beyond tubular anatomy.
>
> ---
>
> **Q4: Formulation is generic but experiments use few covariates.**
>
> We have added a note in **Section 5: Limitations and Future Work**  acknowledging this gap. Our experiments use two (synthetic), three (airway), and four (OASIS) covariates, reflecting the most clinically relevant variables in each setting. The marginalization framework supports arbitrary $N$ with computational cost scaling as $O(LN)$ for the mean. Scaling to larger covariate spaces is noted as future work.

---

> > ### Comment · Reviewer_PF9C · 2026-02-24
> >
> > Thank you for the updates to the manuscript. These changes address the suggestions given by this reviewer. The resulting paper is more clear and looks correct.

---

> > > ### Author Response · Authors · 2026-02-24
> > > **Thanks!**
> > >
> > > Thank you for taking the time to review our revised manuscript. We appreciate your constructive feedback throughout the review process.

---

### Review · Reviewer_MpDr · 2026-02-11

**Summary Of Contributions:**

This paper presents LucidAtlas, a probabilistic framework for learning covariate-conditioned anatomical “atlas” representations that vary over a spatial domain (currently tested for 1D), while also modeling heteroscedastic uncertainty. Building on neural additive model ideas, LucidAtlas decomposes the predicted mean and variance into per-covariate additive components that depend on both the covariate value and spatial location, enabling direct inspection of how each covariate contributes to population trends and variability. A key methodological contribution is the dependence-aware marginalization procedure. LucidAtlas learns an auxiliary covariate dependence model (DeepSets-based) to approximate conditional covariate distributions, which is then used to compute subset-conditioned predictive distributions without retraining. Empirically, the authors validate on a synthetic dataset, OASIS brain volume regression, and a pediatric airway curve dataset; results indicate strong performance on population trend estimation and distributional fit, and improved alignment of covariate interpretations with the observed data distribution when dependence is accounted for.

Key strengths:
* A coherent integration of interpretability, spatial structure, and aleatoric uncertainty in one model with potential high impact
* Rigorous empirically result with clinically relevant evaluations (especially airway OOD detection) with ablations
* The paper is well structured with a clear flow

Key weaknesses/limitations:
* Most of the clinical spatial data are in 2D or even 3D, this might limit the applicability of the method to many clinical problems

**Audience:**

Yes

**Audience Explanation:**

Provide a new way to analyze covariate dependent data, which could be applied to other non-clinical field as well.

**Claims And Evidence:**

Yes

**Claims Explanation:**

* Proper validation for most of the core technical claims: the dependence-aware marginalization is well-motivated and supported by both controlled synthetic evidence and real-data improvements in interpretation alignment.
* Strongest empirical support appears in the pediatric airway setting, where the method’s spatial uncertainty modeling and OOD scoring demonstrate clear gains relative to uncertainty-aware baselines, and ablations suggest monotonic priors and missingness handling help

**Requested Changes:**

* I am curious how would Gaussian process perform on the scalar brain task, this would provide a better baseline to substantiate performance claims in low-dimensional settings.
* Include at least one non-additive synthetic ground truth to test robustness when additive assumption is violated.
* Clarify scope of “atlas” claims by explicitly discussing expected modifications and computational implications for extending from 1D curves to 2D/3D anatomical fields, and what evidence would be needed to support that extension. Be more specific in the abstract section regarding the dataset used.

---

> ### Author Response · Authors · 2026-02-20
> **Response to Reviewer MpDr for LucidAtlas  (Part 1)**
>
> We thank the reviewer for the positive assessment and constructive suggestions. We address each requested addition below.
>
> ---
>
> **Weakness: Most clinical spatial data are in 2D or 3D, limiting applicability.**
>
> We agree that this is the primary limitation of the current validation. To better clarify the scope of our work, we have revised the **Section 1: Introduction** to lead with the clinical application: *tubular anatomies (airways, vessels, intestines) where correspondence is established via centerlines and geometry is summarized as scalar cross-sectional area, yielding a natural 1D spatial setting*. The framework itself does not assume 1D inputs — the subnetworks $f\_i(c\_i, x)$ accept spatial coordinates of arbitrary dimension — but the 1D setting reflects our motivating application rather than an architectural constraint. We have added a detailed discussion in **Section 5: Limitations and Future Work** on extensions to higher-dimensional settings (see Q3 below).
>
> ---
>
> **Q1: GP baseline on the OASIS brain task.**
>
> We have added a Gaussian Process regression baseline (scikit-learn GaussianProcessRegressor, RBF \+ WhiteKernel, same 5-fold cross-validation protocol). Results on the OASIS Brain Volume dataset:
>
> | Method | Pri. | Imp. | SMAPE | NLL |
> | :---- | :---- | :---- | :---- | :---- |
> | LightGBM | ✗ | ✗ | 2.931 ± 0.287 | — |
> | EBM | ✗ | ✗ | 2.889 ± 0.157 | — |
> | NAMESB | ✗ | ✗ | 3.253 ± 0.221 | — |
> | GP | ✗ | ✗ | 2.984 ± 0.227 | **0.576 ± 0.056** |
> | PlainMLP | ✗ | ✗ | 2.888 ± 0.116 | 0.666 ± 0.090 |
> | GAMLSS | ✗ | ✗ | 2.929 ± 0.202 | 0.594 ± 0.039 |
> | NAMLSS | ✗ | ✗ | 2.923 ± 0.192 | 0.736 ± 0.036 |
> | LA-NAM | ✗ | ✗ | 2.959 ± 0.223 | 0.750 ± 0.018 |
> | Ours | ✗ | ✗ | **2.869 ± 0.155** | 0.580 ± 0.046 |
> | Ours | ✗ | ✓ | 2.890 ± 0.178 | 0.577 ± 0.054 |
> | Ours | ✓ | ✗ | 2.883 ± 0.139 | **0.576 ± 0.031** |
> | Ours | ✓ | ✓ | 2.911 ± 0.131 | 0.579 ± 0.038 |
>
> GP achieves competitive NLL but assumes globally homoscedastic noise and provides no per-covariate decomposition. LucidAtlas achieves comparable or better predictive accuracy while additionally providing interpretable covariate effects and heteroscedastic uncertainty estimation.
>
> ---
>
> **Q2: Non-additive synthetic ground truth to test robustness.**
>
> We constructed a non-additive extension of our synthetic dataset with a controlled interaction term: $$y \= f\_1(x, c\_1) \+ f\_2(x, c\_2) \+ \\alpha \\cdot c\_1 c\_2 \\frac{\\sin(0.3x)}{50} \+ \\varepsilon,$$ where $\\alpha \\in \\{0, 0.5, 1.0\\}$ controls interaction strength. We compare $\\texttt{LucidAtlas}$ and  $\\texttt{PlainMLP}$, with and without an explicit interaction feature $c\_3 \= c\_1 \\cdot c\_2$.
>
> The primary results are as follows:
>
> **Robustness to Non-Additive Interactions (Independent $c\_1, c\_2 \\sim U(0,10)$) — MAE ↓**
>
> | Method | $\\alpha=0.0$ | $\\alpha=0.5$ | $\\alpha=1.0$ |
> | :---- | :---: | :---: | :---: |
> | LucidAtlas | 0.160±0.001 | 0.168±0.001 | 0.188±0.000 |
> | PlainMLP | 0.161±0.002 | 0.160±0.001 | **0.160±0.001** |
> | LucidAtlas+C3 | **0.159±0.001** | **0.160±0.000** | 0.160±0.001 |
> | PlainMLP+C3 | 0.160±0.001 | 0.160±0.001 | 0.160±0.001 |
>
> **Robustness to Non-Additive Interactions (Independent $c\_1, c\_2 \\sim U(0,10)$) — NLL ↓**
>
> | Method | $\\alpha=0.0$ | $\\alpha=0.5$ | $\\alpha=1.0$ |
> | :---- | :---: | :---: | :---: |
> | LucidAtlas | \-0.794±0.064 | \-0.736±0.074 | \-0.615±0.040 |
> | PlainMLP | \-0.840±0.009 | \-0.834±0.017 | **\-0.843±0.004** |
> | LucidAtlas+C3 | \-0.758±0.071 | \-0.756±0.086 | \-0.772±0.082 |
> | PlainMLP+C3 | **\-0.841±0.003** | **\-0.839±0.005** | \-0.842±0.005 |
>
>
> Full results and analysis are provided in **Supplementary Sec. S.7**.
>
> As expected, $\\texttt{LucidAtlas}$ without $c\_3$​ degrades as $\\alpha$ increases, while $\\texttt{PlainMLP}$ remains stable. Adding  $c\_3$ ​ as an explicit covariate recovers performance comparable to $\\texttt{PlainMLP}$ in regression metrics (MAE, RMSE). For distribution modeling (NLL), $\\texttt{LucidAtlas+C3}$ also stabilizes across all $\\alpha$ values with only a minor gap compared to $\\texttt{PlainMLP}$. Overall, this demonstrates that the additive limitation can be effectively mitigated by incorporating known interaction features — preserving per-covariate interpretability while accommodating non-additive effects.
>
> ---

---

> > ### Author Response · Authors · 2026-02-20
> > **Response to Reviewer MpDr for LucidAtlas (Part 2)**
> >
> > **Q3: Discuss extension from 1D to 2D/3D; clarify scope of "atlas" claims.**
> >
> > We have added a paragraph in **Section 5: Limitations and Future Work** distinguishing two extension axes: (1) higher-dimensional spatial inputs with scalar outputs (e.g., cortical thickness on brain surfaces) require no modification to the framework; (2) multi-dimensional outputs (e.g., deformation fields) require extending the predictive distribution to a multivariate Gaussian, while the additive decomposition and marginalization carry over directly. Validation on 2D/3D benchmarks such as vertex-wise cortical thickness modeling (Human Connectome Project) or deformation field estimation (e.g., 3D brain meshes or  pediatric airway shapes) is noted as planned future work.
> >
> > ---
> >
> > **Q4: Be more specific about datasets in the Abstract.**
> >
> > Done. The abstract now explicitly names the synthetic spatiotemporal dataset, the pediatric airway shape dataset, and the OASIS brain volume dataset.

---

### Author Response · Authors · 2026-02-20
**Summary of Revisions**

## Summary of Revisions

We thank the Action Editor and all three reviewers for their constructive feedback. Several concerns overlap across reviews; we briefly summarize the key revisions here before addressing each reviewer individually.

**Application-first narrative.**(`PF9C-Q1, qZ92-Q1`) We have restructured **Section 1: Introduction** to lead with the clinical problem — scalar atlas construction for tubular anatomies, with pediatric airway as the primary example — before presenting the technical framework. Design choices in the Method section now include concrete clinical motivation where they are introduced (e.g., monotonicity prior grounded in airway CSA increasing with age; DeepSets motivated by partial covariate availability).

**Scope of "atlas" and 1D limitation** (`PF9C-Q2, MpDr-Weakness, MpDr-Q3`). **Section 1: Introduction** now distinguishes our scalar atlas setting from traditional 2D/3D deformation-based atlases. **Section 5: Limitations and Future Work** adds a discussion on extensions to higher-dimensional settings, clarifying that the 1D setting reflects our motivating application, not an architectural constraint.

**Spatial dependence modeling.** (`qZ92-Q1, qZ92-Q2`) A new paragraph in **Section 2: Related Work** positions $\\texttt{LucidAtlas}$ relative to spatial random effect approaches (e.g., Gaussian processes for continuous domains (Rasmussen, 2003), CAR models (Banerjee et al., 2003\) for discrete areal data). These methods capture spatial correlations but treat covariate effects as global fixed effects — the same at every location. ***$\\texttt{LucidAtlas}$ models the nonlinear interaction between covariates and spatial location via $f\_i(c\_i, x)$, enabling location-specific covariate effects (e.g., how age influences CSA differently at the subglottis vs. carina).***

**Dataset roles .** `(PF9C-Q3, MpDr-Q4)` **Section 4.1: Datasets & Experimental Protocols** now explicitly describes each dataset's role: synthetic for ground-truth validation, pediatric airway as the primary application, OASIS as a generality check.

**New experiments.** `(MpDr-Q1, MpDr-Q2)` We added a GP baseline on OASIS  and a non-additive synthetic experiment (Supplementary Sec. S.7).

All changes are highlighted in ***blue*** in the revised manuscript.

\[1\] Banerjee, Sudipto, Bradley P. Carlin, and Alan E. Gelfand. *Hierarchical modeling and analysis for spatial data*. Chapman and Hall/CRC, 2003\.
\[2\] Rasmussen, Carl Edward."Gaussian processes in machine learning." *Summer school on machine learning*. Berlin, Heidelberg: Springer Berlin Heidelberg, 2003\. 63-71.

---

### Decision · Action_Editor_kY8i · 2026-03-10

**Recommendation:** Accept as is

**Audience:**

Yes

**Audience Explanation:**

Yes: the paper introduces a way to build covariate-dependent biological atlases, which may be of interest to machine learning researchers who work in biomedical application areas. There are also clear avenues for future development, especially with regards to extensions to higher-dimensional atlases

**Claims And Evidence:**

Yes

**Claims Explanation:**

The reviewers and I agree that the paper presents a well-conducted study, explains the methodology well, and clearly substantiates its claims. The revised manuscript is also much clearer.